# OPENSIR: OPEN-ENDED SELF-IMPROVING REASONER

## ABSTRACT

Recent advances in large language model (LLM) reasoning through reinforcement learning rely on annotated datasets for verifiable rewards, which may limit models' ability to surpass human-level performance. While self-play offers a promising alternative, existing approaches depend on external verifiers or cannot learn open-endedly. We present **Open**-Ended **S**elf-**I**mproving **R**easoner (OpenSIR), a self-play framework where an LLM learns to generate and solve novel problems by alternating teacher and student roles without external supervision. To generate novel problems, OpenSIR optimises for both difficulty and diversity, rewarding problems that challenge appropriately while exploring distinct concepts, enabling open-ended mathematical discovery. Starting from a single trivial seed problem, OpenSIR substantially improves instruction models: Llama-3.2-3B-Instruct advances from 73.9 to 78.3 on GSM8K, and from 28.8 to 34.4 on College Math, while Gemma-2-2B-Instruct rises from 38.5 to 58.7 on GSM8K. Our analyses reveal that OpenSIR achieves open-ended learning through co-evolving teacher-student roles that adaptively calibrate difficulty and drive diverse exploration, progressing autonomously from basic to advanced mathematics.

## 1 INTRODUCTION

Reinforcement learning with verifiable rewards (RLVR) drives recent advances in LLM reasoning. Recent works on DeepSeek-R1 (DeepSeek-AI et al., 2025) and OpenAI o1 (OpenAI, 2024) have shown that large-scale reinforcement learning improves reasoning capabilities. Yet, these methods require extensive human-annotated data for reward signals, which bottleneck scalability and potentially limit performance to human-level (Hughes et al., 2024b).

One promising direction to address these fundamental limitations is to generate synthetic training data through self-play, which demonstrated remarkable success in various games (Silver et al., 2016; 2017; Brown & Sandholm, 2019; FAIR et al., 2022), allowing systems to exceed human-level performance by learning from unambiguous reward signals (Silver et al., 2017; FAIR et al., 2022). Yet, mathematical reasoning poses a key challenge for self-play: unlike games that have clear rules and winners, generated mathematics problems lack the ground-truth answers to provide feedback signals. Recent works utilise external verifiers, such as compilers for coding tasks (Pourcel et al., 2024; Zhao et al., 2025) or game rules (Liu et al., 2025), while R-Zero (Huang et al., 2025) employs majority voting with basic repetition penalties. However, these approaches cannot achieve open-ended learning, the ability to continuously generate and pursue novel challenges without external supervision (Bauer et al., 2023; Hughes et al., 2024a), confining systems to known concepts instead of exploring diverse mathematical domains.

We present **Open**-ended **S**elf-**I**mproving **R**easoner (OpenSIR), a method for training a policy $\pi_\theta$ to generate and solve novel problems without external supervision. OpenSIR uses *self-play* — a single policy $\pi_\theta$ alternates between teacher and student roles: the teacher generates problems, while the student solves them, with problem-solution pairs selected for reinforcement learning updates. We reward teachers for generating appropriately challenging problems for the students, using consistency and solution length across multiple solution attempts. OpenSIR achieves open-ended learning through embedding-based diversity rewards that drive continuous exploration of novel mathematical concepts.

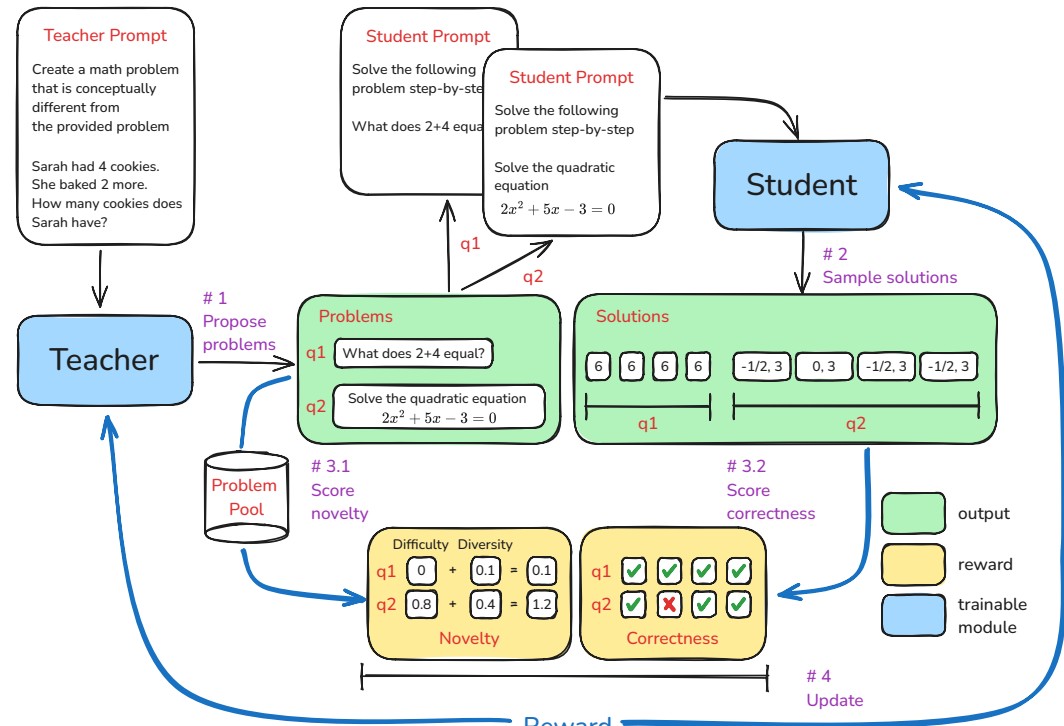

Figure 1: Overview of the OpenSIR framework. A single policy $\pi_\theta$ alternates between generating and solving novel problems without external supervision. Each training iteration consists of **problem generation**, **solution sampling**, **scoring**, and **model update**. Novelty is captured through both *difficulty* and *diversity*: problems must be challenging yet solvable, and they must explore new concepts. These dimensions together drive open-ended self-improvement in the LLM reasoning ability.

Our experiments show OpenSIR outperforms base instruction models and reinforcement learning baselines. Starting from a single trivial seed problem, OpenSIR improves base instruction models by up to 6.3 accuracy points, surpassing GRPO baselines trained on thousands of human-annotated examples. Specifically, Llama-3.2-3B-Instruct improves from 73.9→78.3 (+4.4) on GSM8K and 28.8→34.4 (+5.6) on College Math, while Gemma-2-2B-Instruct rises from 38.5→58.7 (+20.2) on GSM8K and 19.1→23.4 (+4.3) on College Math.

Our qualitative analysis reveals OpenSIR succeeds through adaptive difficulty calibration and diversity-driven exploration. Problem difficulty is automatically calibrated throughout training, while the range of topics expands from basic to advanced mathematics (§4.1). Generating harder problems risks invalidity, requiring a balance between challenge and correctness (§4.2). Diversity rewards incentives generate problems spanning varied mathematical concepts (§4.3). Teacher-student co-evolution proves essential: without teacher training, models cannot generate appropriate challenges or explore new topics (§4.4).

## 2 OPEN-ENDED SELF-IMPROVING REASONER

Figure 1 illustrates the Open-Ended Self-Improving Reasoner (OpenSIR), a self-play framework in which a policy $\pi_\theta$ learns to both generate and solve novel mathematical problems without external supervision. We use reinforcement learning to optimise two roles within one policy: the *teacher*, which creates new problems, and the *student*, which solves them. This open-ended approach enables the policy to bootstrap its learning and discover new and diverse challenges without annotated data. Each training iteration involves four phases:

1. **Problem generation** (§2.1): The teacher proposes new problems by conditioning on reference problems from an accumulated pool of previously generated problems;

2. **Solution sampling** (§2.2): The student attempts multiple solutions per problem, with majority voting determining the reference answer and solve rate measuring reliability;

3. **Scoring** (§2.3): We compute novelty scores for the teacher's generated problems and correctness scores for the student's solutions; and

4. **Model update** (§2.4): We update the policy's parameters with role-specific rewards using the problem-solution pairs selected by the novelty scores.

Algorithm 1 summarizes the complete training procedure.

In OpenSIR, we define novelty along two dimensions that together drive continuous open-ended learning. First, problems must have an appropriate level of difficulty. It should be challenging enough to promote learning but solvable enough to provide reliable training signals. Second, problems must explore diverse concepts, preventing the model from repeating learning on familiar concepts. This two-dimensional view of novelty ensures the model continuously expands both the depth and breadth of its mathematical reasoning abilities.

## 2.1 PROBLEM GENERATION

At each iteration $t$, the policy $\pi_\theta$ generates $k$ groups of $G$ problems each, denoted as $q_{1:G}$ within each group, for a total of $M = k \times G$ problems. To generate these problems, we sample $k$ reference problems from a pool $\mathcal{P}_{t-1}$ of accumulated problems from previous iterations, where each reference problem serves as a seed for generating $G$ new problems. Each generated problem must explicitly include the mathematical concepts required for its solution. Problems with invalid formats are filtered out, and valid problems proceed to the solution-sampling phase. We initialise the problem pool $\mathcal{P}_0$ with a single trivial problem ("What is 1+1?").

## 2.2 SOLUTION SAMPLING

Let $a_j$ denote the parsed answer from solution attempt $o_j$. We select the most common answer across attempts as the reference answer $a^*$. We then compute the *solve rate* for each problem to determine the reliability of the answers. For brevity, we denote $s_{q_i} = \text{SolveRate}(q_i)$ when referring to the solve rate of problem $q_i$.

$$\text{SolveRate}(q_i) = \frac{\text{count}(a^*)}{G} \quad \text{where} \quad a^* = \underset{a \in a_{1:G}}{\arg\max} \text{count}(a), \tag{1}$$

In Eq. (1), $\text{count}(a)$ denotes the number of times answer $a$ appears. The solve rate quantifies answer reliability. High solve rates indicate reliable reference answers due to solution convergence, while low solve rates suggest inconsistent solutions that may indicate flawed problem formulations.

## 2.3 SCORING

We evaluate the quality of generated problems and solutions with different scoring functions. The teacher's problems are scored based on *difficulty* and *diversity*, while the student's receive scores for *correctness*. Additionally, both roles incorporate format scores to ensure parseable outputs.

### 2.3.1 TEACHER SCORING

We capture novelty through two fundamental dimensions: difficulty and diversity. We measure difficulty using *solvability* to ensure problems remain appropriately challenging and *solution length* to encourage multi-step reasoning, as these provide complementary signals about problem difficulty. Diversity is promoted through embedding distance, which encourages exploration of varied mathematical concepts. These components form a unified novelty score that guides problem generation.

**Solvability (score$_{\text{sol}}$).** The solvability score identifies problems with appropriate challenge. We use solve rate as a proxy for solvability—problems with $s_{q_i} > s_{\max}$ are likely too easy, while those with $s_{q_i} < s_{\min}$ are either too difficult or malformed. We employ a triangular scoring function that peaks at the optimal solve rate and decreases linearly as problems become too easy or too hard.

We define the solve rate range as $[s_{\min}, s_{\max}]$. Easy problems ($s_{q_i} > s_{\max}$) fail to challenge the model, while problems that are too hard or malformed ($s_{q_i} < s_{\min}$) offer minimal training value.

Formally, for $s_{q_i} \in [0, 1]$, let $s_{\mathrm{mid}} = (s_{\min} + s_{\max})/2$ be the midpoint:

$$\text{score}_{\mathrm{sol}}(q_i) = \begin{cases} 1 - \alpha|s_{q_i} - s_{\mathrm{mid}}| & \text{if } s_{q_i} \in [s_{\min}, s_{\max}], \\ 0 & \text{otherwise} \end{cases} \tag{2}$$

where $\alpha = (1 - 1/G)/(s_{\mathrm{mid}} - s_{\min})$ is the slope coefficient, with $G$ being the number of solution attempts. The score peaks at the midpoint $s_{\mathrm{mid}}$ and decreases to $1/n$ at the boundaries.

This creates a symmetric triangular score centred at the midpoint of the solve rate range, giving a maximum score for problems with moderate difficulty and progressively less score as the solve rate approaches either boundary.

**Solution Length ($\text{score}_{\mathrm{len}}$).** Solution length complements solvability by measuring problem complexity. Problems requiring multi-step reasoning typically elicit longer solutions. We score problems using the average length of student solutions:

$$\text{score}_{\mathrm{len}}(q_i) = \min\left(\frac{\bar{l}(q_i)}{l_{\mathrm{base}}}, \frac{l_{\mathrm{cap}}}{l_{\mathrm{base}}}\right) \tag{3}$$

where $\bar{l}(q_i)$ denotes average solution length for problem $q_i$, $l_{\mathrm{base}}$ is a normalisation factor (defaults to 1000 tokens), and $l_{\mathrm{cap}}$ prevents outliers from dominating the scoring signal. This score complements the solvability score (see Appendix C.1).

**Diversity ($\text{score}_{\mathrm{div}}$).** We compute the semantic distance between each new problem and the existing problem pool:

$$\text{score}_{\mathrm{div}}(q_i) = \min_{q' \in \mathcal{P}_{t-1}} d(e_{q_i}, e_{q'}) \tag{4}$$

where $e_{q_i}$ and $e_{q'}$ represent problem embeddings obtained from a pre-trained encoder, and $d(\cdot, \cdot)$ denotes cosine distance. This score maximises when a problem is semantically distant from all existing problems in the pool.

**Format ($\text{score}_{\mathrm{fom}}^T$).** The format score ensures proper problem structure. Generated problems must be enclosed in <question> tags with concepts listed in <concepts> tags (maximum three concepts). We assign $\text{score}_{\mathrm{fom}}^T(q_i) = 1$ for correct formatting and $\text{score}_{\mathrm{fom}}^T(q_i) = 0$ otherwise.

**Novelty Score.** We combine these components into a novelty score capturing both difficulty and diversity:

$$\text{score}_{\mathrm{novel}}(q_i) = \alpha\text{score}_{\mathrm{sol}}(q_i) + \lambda\text{score}_{\mathrm{len}}(q_i) + \gamma\text{score}_{\mathrm{div}}(q_i) + \delta\text{score}_{\mathrm{fom}}^T(q_i) \tag{5}$$

where $\alpha$, $\lambda$, $\gamma$, $\delta$ are hyperparameters that control the relative importance of each component. This novelty score is used to select high-quality problem-solution pairs for training.

### 2.3.2 STUDENT SCORING

The student's score is based on solution correctness. For each solution attempt, we evaluate correctness by comparing the parsed answer against the reference answer from majority voting.

**Format ($\text{score}_{\mathrm{fom}}^S$).** The format score ensures proper answer presentation. Solutions must present final answers in \boxed{} notation. We assign $\text{score}_{\mathrm{fom}}^S(o_j) = 1$ for correct formatting and 0 otherwise.

**Correctness Score.** The student's correctness score combines accuracy with the format score:

$$\text{score}_{\mathrm{correct}}(o_j, a_j) = \mathbf{1}[a_j = a^*] + \delta\text{score}_{\mathrm{fom}}^S(o_j) \tag{6}$$

where $\mathbf{1}[a_j = a^*]$ is an indicator function that equals 1 when parsed answer $a_j$ from outcome $o_j$ matches the reference answer $a^*$, and 0 otherwise. This correctness score evaluates both solution accuracy and proper formatting.

---

**Algorithm 1** OpenSIR

---

**Require:** Problem pool $\mathcal{P}_0$, policy $\pi_\theta^{(0)}$, embedding encoder $\varepsilon$, batch size $B$, generation group size $G$, solve rate range $[s_{\min}, s_{\max}]$, teacher prompt $p_T$, student prompt $p_S$
1: **for** $t = 1$ to $T$ **do**
2:     Sample $k = B/G$ reference problems $\{p_1, \ldots, p_k\}$ from $\mathcal{P}_{t-1}$                $\triangleright$ Problem Generation
3:     **for** $i = 1$ to $k$ **do**
4:         Sample $q_{i,1:G} \sim \pi_\theta^{(t)}(\cdot \mid p_i, p_T)$
5:     **end for**
6:     $\mathcal{Q}_{\text{valid}} \leftarrow \{q_{i,j} \mid q_{i,j} \text{ has valid format}\}$
7:     **for** each $q_i \in \mathcal{Q}_{\text{valid}}$ **do**                           $\triangleright$ Solution Sampling
8:         Sample solutions $o_{i,1:G} \sim \pi_\theta^{(t)}(\cdot \mid q_i, p_S)$
9:         Parse answers $a_{i,1:G}$ from solutions $o_{i,1:G}$
10:        Compute reference answer $a_i^* = \arg\max_{a \in a_{i,1:G}} \text{count}(a)$ via majority voting
11:        Compute solve rate $s_{q_i} = \text{count}(a_i^*)/G$
12:        Compute embedding $e_{q_i} \leftarrow \varepsilon(q_i)$
13:     **end for**
14:     Compute $\text{score}_{\text{novel}}(q_i)$ for all $q_i \in \mathcal{Q}_{\text{valid}}$ via Eq. 5            $\triangleright$ Scoring
15:     $\mathcal{I}_T \leftarrow \text{top}_{B/(2G)}(i : \text{Var}(\text{score}_{\text{novel}}(q_{i,1:G})), i \in \{1, \ldots, k\})$     $\triangleright$ Teacher sample selection
16:     $\mathcal{Q}_S \leftarrow \text{top}_{B/(2G)}(q : \text{score}_{\text{novel}}(q), q \in \mathcal{Q}_{\text{valid}})$        $\triangleright$ Student sample selection
17:     Compute $\text{score}_{\text{correct}}(o_{i,j}, a_{i,j})$ for solutions where $q_i \in \mathcal{Q}_S$ via Eq. 6
18:     $\mathcal{D}_T \leftarrow \{(p_T, q_{i,j}, R_{i,j}^T) : i \in \mathcal{I}_T, 1 \leq j \leq G\}$ where $R_{i,j}^T = \text{score}_{\text{novel}}(q_{i,j})$     $\triangleright$ Model Update
19:     $\mathcal{D}_S \leftarrow \{(p_S, o_{i,j}, R_{i,j}^S) : q_i \in \mathcal{Q}_S, 1 \leq j \leq G\}$ where $R_{i,j}^S = \text{score}_{\text{correct}}(o_{i,j}, a_{i,j})$
20:     Update $\pi_\theta^{(t+1)} \leftarrow \text{GRPO}(\pi_\theta^{(t)}, \mathcal{D}_T \cup \mathcal{D}_S)$
21:     $\mathcal{P}_t \leftarrow \mathcal{P}_{t-1} \cup \mathcal{Q}_{\text{valid}}$
22: **end for**
23: **return** $\pi_\theta^{(T)}$

---

## 2.4 Model Update

After computing novelty scores, we select $B$ high-quality samples from valid problems for reinforcement learning, allocating half to problem generation and half to solution solving. For teacher training, we choose problem groups with highest $\text{score}_{\text{novel}}$ variance to ensure diverse training signals. For student training, we select problems with the highest novelty scores to provide maximal training value.

We optimise the policy using $\pi_\theta$ with an objective similar to Group Relative Policy Optimization (GRPO) (Shao et al., 2024), adapted for on-policy training to ensure stability (Chen et al., 2025):

$$\mathcal{J}(\theta) = \mathbb{E}_{\substack{q_{1:G} \sim \pi_\theta(\cdot|p_T) \\ o_{1:G} \sim \pi_\theta(\cdot|q_i, p_S)}} \left[ \sum_{r \in \{T,S\}} \frac{1}{G} \sum_{i=1}^{G} A_i^r \right] - \beta \mathbb{D}_{KL}\left(\pi_\theta \| \pi_{\text{ref}}\right) \tag{7}$$

where $p_T$ and $p_S$ are the teacher and student prompts respectively, $r \in \{T, S\}$ refers to teacher and student, $\mathbb{D}_{KL}$ denotes the KL divergence, $\pi_{\text{ref}}$ refers to the initial model before training. The advantage for each role $r \in \{T, S\}$ is computed as:

$$A_i^r = \frac{R_i^r - \text{mean}\left(R_{1:G}^r\right)}{\text{std}\left(R_{1:G}^r\right)}. \tag{8}$$

We define role-specific rewards $R_i^T$ and $R_j^S$ using the scoring functions from Section 2.3:

$$R_i^T = \text{score}_{\text{novel}}(q_i), \quad R_j^S = \text{score}_{\text{correct}}(o_j, a_j) \tag{9}$$

All valid problems are then added to the problem pool $\mathcal{P}_t$ for future iterations.

## 3 Experiments

### 3.1 Training Setup

We experiment with four instruction-tuned models: Llama-3.2-3B-Instruct, Llama-3.1-8B-Instruct (Dubey et al., 2024), Gemma-2-2B-Instruct (Team et al., 2024), and Qwen-2.5-3B-Instruct (Team,

2024) with GRPO (Shao et al., 2024). We use a learning rate of $3 \times 10^{-7}$ and 10 warm-up steps. The KL divergence coefficient is set to $10^{-4}$ and the batch size is 256. To compare models trained on the same number of problem-solution pairs, we train the GRPO baselines with 100 steps, and OpenSIR for 200 steps since OpenSIR allocates half of its training budget to problem generation. Clipping is not applied since we strictly use on-policy samples. Each experiment is run with three random seeds. We provide full training details in Appendix D.1.

## 3.2 DATASET AND EVALUATION SETUP

We evaluate method on five mathematical benchmarks: GSM8K (Cobbe et al., 2021), MATH-500 (Hendrycks et al., 2021), Minerva (Lewkowycz et al., 2022), OlympiadBench (He et al., 2024), and College Math (Tang et al., 2024).

We use sampling temperature 0.6 and top-p 0.95. The maximum response length is set to 4,096 tokens. We report the average performance over 16 generations (avg@16). Answer extraction and comparison are performed using the `math_verify` library.

## 3.3 BASELINES

(1) **Base** We evaluate the instruction-tuned models using zero-shot prompting, where models generate step-by-step reasoning and provide final answers without additional training.

(2) **GRPO** We train the instruction models with GRPO (Shao et al., 2024) on established mathematical datasets. We train two variants: **GRPO$_{math}$** on the MATH dataset (7,500 training examples) (Hendrycks et al., 2021) and **GRPO$_{gsm8k}$** on the GSM8K dataset (7,473 training examples) (Cobbe et al., 2021).

(3) **Absolute Zero** (Zhao et al., 2025) A self-play framework for code generation that uses Python as external verifier, rewarding problems with minimal solve rates.

(4) **R-Zero** (Huang et al., 2025) A verifier-free self-play framework that trains separate challenger and solver models using rewards based on repetition penalties and solve rates near 0.5. OpenSIR differs by explicitly optimising for diversity and incorporating solution length to capture multiple dimensions of difficulty within a single model.

## 3.4 MAIN RESULTS

Table 1 demonstrates that OpenSIR achieves substantial gains over the base instruction models across different model scales and families. OpenSIR improves Llama-3.2-3B-Instruct by 3.6 points, Llama-3.1-8B-Instruct by 3.1, and Gemma-2-2B-Instruct by 6.3 points on average accuracy. One exception is Qwen-2.5-3B-Instruct (+.0.6), where all methods show limited gains. The limited improvement aligns with observations of potential benchmark contamination (Wu et al., 2025).

OpenSIR outperforms all GRPO baselines without using human-annotated training data. GRPO baselines require over 7,000 labeled examples, yet OpenSIR generates its own training problems through self-play, starting from a single trivial seed problem. OpenSIR also substantially outperforms other self-play methods by 1.75 to 3.38 points on Llama-3.2-3B-Instruct, Gemma-2-2B-Instruct, and Llama-3.1-8B-Instruct. Although Absolute Zero and R-Zero demonstrate significant improvements over non-instruction-tuned models in their original work (Zhao et al., 2025; Huang et al., 2025), both show limited gains on instruction-tuned models. This challenging scenario is precisely where self-play methods are intended to excel - when models have already consumed available human-annotated data. As we reveal later in our analysis, the success of OpenSIR can be attributed to its ability to explore diverse mathematical concepts, and calibrating difficulty adaptively to maintain optimal challenge levels (§4.1). These capabilities enable OpenSIR to self-improve and expand its skills without external training data, achieving open-ended learning.

## 4 ABLATIONS AND ANALYSES

We perform a series of ablation studies and qualitative analyses on Llama-3.2-3B-Instruct to dissect the contribution of each key component in the OpenSIR framework. Our analysis investigates:

| Model | GSM8K | MATH-500 | Minerva | College Math | OlympiadBench | Avg. |
|---|---|---|---|---|---|---|
| **Llama-3.2-3B-Instruct** | | | | | | |
| Base | 73.94 | 42.86 | 15.21 | 28.78 | 13.09 | 34.78 |
| GRPO$_{gsm8k}$ | **79.72** | 45.30 | 16.27 | 33.33 | 14.56 | 37.83$^{+3.05}$ |
| GRPO$_{math}$ | 76.48 | 45.26 | 16.09 | 32.95 | 14.13 | 36.98$^{+2.20}$ |
| Absolute Zero | 74.37 | 44.71 | 14.78 | 31.93 | 14.42 | 36.04$^{+1.26}$ |
| R-Zero | 76.34 | 44.27 | 15.84 | 32.72 | 14.19 | 36.67$^{+1.89}$ |
| OpenSIR | 78.28 | **46.22** | **17.46** | **34.42** | **15.72** | **38.42**$^{+3.64}$ |
| **Gemma-2-2B-Instruct** | | | | | | |
| Base | 38.50 | 16.51 | **10.09** | 19.11 | 3.00 | 17.44 |
| GRPO$_{gsm8k}$ | **58.75** | 19.15 | 7.75 | 20.45 | 3.21 | 21.86$^{+4.42}$ |
| GRPO$_{math}$ | 56.03 | 22.76 | 7.96 | 16.31 | **3.24** | 21.26$^{+3.82}$ |
| Absolute Zero | 57.13 | 15.92 | 8.29 | 17.36 | 3.18 | 20.38$^{+2.94}$ |
| R-Zero | 56.37 | 17.31 | 8.49 | 19.86 | 3.12 | 21.03$^{+3.59}$ |
| OpenSIR | 58.03 | **24.75** | 9.51 | **23.36** | 3.15 | **23.76**$^{+6.32}$ |
| **Qwen-2.5-3B-Instruct** | | | | | | |
| Base | 84.43 | 65.36 | 25.23 | 48.22 | 27.94 | 50.24 |
| GRPO$_{gsm8k}$ | 84.94 | 65.77 | 25.31 | 48.46 | 28.31 | 50.56$^{+0.32}$ |
| GRPO$_{math}$ | 84.31 | **65.89** | 24.98 | 48.34 | 28.26 | 50.36$^{+0.12}$ |
| Absolute Zero | 84.62 | 65.33 | 25.21 | 48.31 | 28.12 | 50.32$^{+0.08}$ |
| R-Zero | 84.22 | 64.93 | 24.81 | 48.45 | 27.82 | 50.05$^{-0.19}$ |
| OpenSIR | **85.38** | 65.87 | **25.96** | **48.74** | **28.33** | **50.85**$^{+0.61}$ |
| **Llama-3.1-8B-Instruct** | | | | | | |
| Base | 84.50 | 47.89 | 22.75 | 34.10 | 16.26 | 41.10 |
| GRPO$_{gsm8k}$ | **88.70** | 50.37 | 24.83 | 35.03 | 16.43 | 43.05$^{+1.95}$ |
| GRPO$_{math}$ | 86.23 | 50.82 | 23.98 | 34.93 | 16.54 | 42.50$^{+1.40}$ |
| Absolute Zero | 86.89 | 51.38 | 23.21 | 34.39 | 15.96 | 42.37$^{+1.27}$ |
| R-Zero | 86.19 | 50.93 | 24.11 | 32.93 | 15.66 | 41.96$^{+0.86}$ |
| OpenSIR | 87.30 | **52.38** | **27.29** | **36.29** | **17.81** | **44.21**$^{+3.11}$ |

Table 1: The avg@16 performance on five mathematical benchmarks. OpenSIR outperforms GRPO baselines trained on >7,000 human-annotated examples and other self-play methods (Absolute Zero, R-Zero) across model families, starting from a single trivial seed problem.

(1) the evolution of problem difficulty and diversity over training (§4.1), (2) the effect of solve rate thresholds on the difficulty-validity trade-off (§4.2), (3) the impact of diversity rewards on promoting exploration of novel problem types (§4.3), and (4) the necessity of dual-role training (§4.4).

## 4.1 EVOLUTION OF PROBLEM DIFFICULTY AND DIVERSITY

We track how difficulty and diversity evolve during training through human evaluation. We sample 20 problems from three OpenSIR training checkpoints (steps 0, 100, 200) and 20 each from GSM8K and MATH. Annotators evaluate mixed sets of five problems (one per source), identifying topics, assessing validity, and ranking difficulty. Figure 2 shows average difficulty rankings (1=easiest, 5=hardest); see Appendix B for full annotation instructions.

Figure 2 (left) reveals a V-shaped difficulty trend across training stages. Problems start at 3.4 difficulty, drop to 3.0 at midpoint, then rise to 3.8. This pattern reflects OpenSIR's self-calibration: the model first generates overly difficulty problems, then learns appropriate difficulty, and finally increases challenge as its solving capabilities improve. The model also generates increasingly valid problems during training — validity improves from below 50% initially to 95% (19 of 20 problems) by the end.

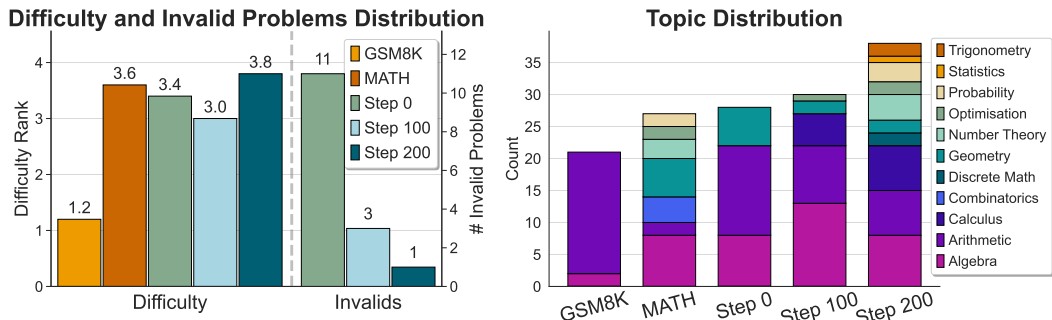

Figure 2: Evolution of problem difficulty, validity, and topic diversity during OpenSIR training. **(Left)** Human evaluation results showing difficulty rankings (1-5 scale where 1=easiest, 5=hardest) and number of invalid problems for GSM8K, MATH, and problems generated at steps 0, 100, and 200 of training. Invalid problems are those with logical flaws, missing information, or ambiguities. **(Right)** Distribution of mathematical topics across training stages, demonstrating the increasing diversity of generated problems from step 0 to step 200.

Figure 2 (right) shows topic diversity expansion across training. OpenSIR progresses from basic topics (algebra, arithmetic, geometry) to advanced domains including calculus and optimisation, eventually incorporating trigonometry, statistics, and other mathematical areas. This progression demonstrates OpenSIR's capacity for autonomous exploration of diverse mathematical concepts. Appendix A.2 provides detailed case studies that illustrate this evolution.

## 4.2 DIFFICULTY-VALIDITY TRADE-OFF

| Model | Acc | Validity | Solve Rate |
|---|---|---|---|
| OpenSIR$_{0.5}$ | **38.42** | 70.82 | 89.82 |
| OpenSIR$_{0.3}$ | 36.81 | 52.32 | 81.38 |
| OpenSIR$_{0.1}$ | 35.97 | 42.31 | 78.31 |

Table 2: Performance, problem validity, and solve rate across different lower solve-rate thresholds, with the upper threshold fixed at 0.9 for all variants. Validity and solve rate are estimated using GPT-5. Lower thresholds produce harder problems but significantly more invalid ones, ultimately reducing overall performance.

We investigate the difficulty-validity trade-off by training OpenSIR variants with lower solve-rate thresholds of 0.1, 0.3, and 0.5, keeping the upper threshold at 0.9 From each variant, we sample 300 problems and assess quality with GPT-5 (OpenAI, 2025a) using 8 responses per problem. We measure validity by comparing GPT-5's majority answer to our reference answer and difficulty by GPT-5's solve rate.

Table 2 reveals a clear trade-off between validity and difficulty. While lowering the threshold from 0.5 to 0.1 produces moderately harder problems (GPT-5 solve rate decreases from 89.82% to 78.31%), validity plummets from 70.82% to 42.31%. This suggests that problems with very low solve rates frequently contain errors rather than representing genuine mathematical challenges. performance consistently drops with lower thresholds, supporting our selection of 0.5 as the lower threshold for the solvability reward.

Besides solve-rate thresholds, we find that rewarding longer solutions provides another mechanism for promoting problem complexity that encourage sophisticated multi-step problems (Appendix C.1).

## 4.3 IMPACT OF DIVERSITY REWARDS

We analyse the impact of the diversity reward on problem diversity through problem embeddings, n-gram similarity, and concept overlap. Figure 3 visualises the problem embeddings with t-SNE, where red points represent problems without diversity reward, cyan points show problems with diversity reward, gold indicates MATH dataset problems, and purple marks GSM8K dataset problems. Without diversity rewards, problems cluster in narrow regions, generating similar types repeatedly and failing to achieve open-ended exploration. With diversity rewards, problems spread across the embedding space, reaching areas beyond MATH and GSM8K training sets. Further analysis of n-gram similarity and concept overlap support these findings, demonstrating consistent patterns of greater dispersion and novelty (Appendix A.3).

Table 3 empirically confirms the importance of diversity rewards, showing that removing diversity rewards reduces average performance by 1.97 (from 38.42 to 36.45). It also shows that the number of unique concepts has dropped significantly (from 5914 to 3328). This demonstrates that without diversity rewards, the model generates repetitive problems with limited learning value, constraining the teacher's ability to present varied mathematical challenges to the student. Incorporating diversity rewards thus enables exploration of novel problems beyond existing datasets, supporting open-ended learning where the model continuously discovers new challenges rather than repeating known concepts. Notably, this improvement is robust to the choice of diversity metric (Appendix C.2), with different measurement approaches yielding comparable results.

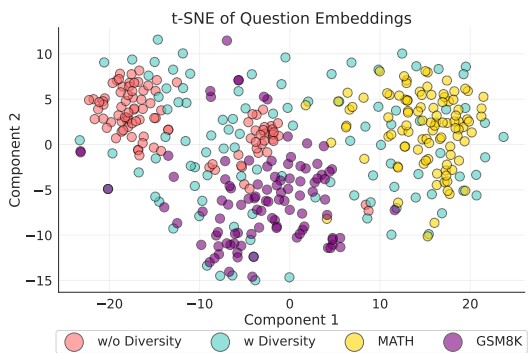

Figure 3: t-SNE visualization of problem embeddings showing the effect of diversity reward on problem distribution. With diversity reward, problems explore broader regions of the embedding space compared to the clustered distribution without diversity reward.

| Model | Acc | # Concepts |
|---|---|---|
| w diversity | **38.42** | 5914 |
| w/o diversity | 36.45 | 3328 |

Table 3: OpenSIR performance with and without diversity reward. Exploring diverse mathematical concepts through the diversity reward improves both accuracy and concept coverage, showing that variety in problem types is crucial for self-improvement.

### 4.4 IMPORTANCE OF DUAL-ROLE TRAINING

| Trained Roles | Acc | Avg. Solve Rate |
|---|---|---|
| Both | **38.42** | 72.20 ($\pm$4.49) |
| Student | 35.89 | 64.56 ($\pm$17.37) |

Table 4: Accuracy and average solve rate with standard deviation ($\pm$) for OpenSIR with teacher training (Both) versus without teacher training (Student only). Joint training achieves higher accuracy and remarkably stable problem difficulty (much lower solve rate variance), demonstrating that teacher training enables calibrated problem generation at optimal difficulty levels for effective learning.

We evaluate the contribution of the joint teacher-student training by testing a variant where only the student is updated while the teacher remains fixed at its initial state. Table 4 shows that accuracy drops significantly from 38.42 to 35.89 when only the student is trained. This demonstrates that effective self-play requires both components to co-evolve.

Without teacher training, generated problems become harder (solve rate drops from 72.20 to 64.56) and drift from the optimal 70% target solve rate established in Section 4.2. More critically, solve rate variance increases tremendously (from $\pm$4.49 to $\pm$17.37), indicating highly inconsistent diffi-

culty during training. This poorly calibrated curriculum explains the performance drop: the fixed teacher cannot adapt to the student's evolving capabilities, whereas joint training enables continuous difficulty calibration at the optimal challenge level.

## 5 RELATED WORK

**Self-play.** Self-play achieved superhuman performance in games without human data, from AlphaGo (Silver et al., 2016; 2017), StarCraft II (Vinyals et al., 2019), Poker (Brown & Sandholm, 2019), DotA (OpenAI et al., 2019), and Diplomacy (FAIR et al., 2022). Baker et al. (2019) show that agents can discover complex strategies with self-play, suggesting it is a promising avenue for continuous open-ended learning. Recent works apply self-play to LLM reasoning: Absolute Zero (Zhao et al., 2025) and Spiral (Liu et al., 2025) rely on external verifiers or game rules that limit their use beyond specific domains. R-Zero (Huang et al., 2025) attempts verifier-free self-play but uses only repetition penalties without a mechanism to encourage exploration, constraining open-ended learning. In contrast, OpenSIR generates and solves problems without external supervision while actively promoting diversity to enable continuous discovery of novel mathematical concepts.

**Reinforcement Learning with Verifiable Feedback (RLVF).** RLVF drives recent advances in LLM reasoning (OpenAI, 2024; 2025b; DeepSeek-AI et al., 2025) but requires extensive human-annotated data for verifiable reward signals (Zeng et al., 2025), creating scalability bottleneck and potentially limiting performance to human-level. Recent works show that moderate-difficulty training samples provide optimal learning signals (Zheng et al., 2025; Sun et al., 2025), while diverse problem types enhance mathematical reasoning (Akter et al., 2025; Chen et al., 2025). These insights directly motivate OpenSIR to optimise for appropriate difficulty calibration and diversity-driven exploration, enable models to learn math reasoning open-endedly without human supervision.

**Synthetic Data.** Synthetic data generation has emerged as a crucial technique for improving mathematical reasoning in LLMs (Yu et al., 2023; Li et al., 2024; Lu et al., 2024). Recent studies have investigated LLM-based problem generation (Toshniwal et al., 2024; Chan et al., 2024; Leang et al., 2025; Havrilla et al., 2025). Diversity has been known to be a critical aspect of synthetic data generation (Havrilla et al., 2024). To enhance diversity, Jung et al. (2025) enhance diversity by measure diversity of data in loss gradients, and Chan et al. (2024) incorporate varied personas in generation prompts, while Havrilla et al. (2025) prioritise problems with diverse skill coverage. Beyond synthetic problems, models can also generate synthetic reasoning traces for self-improvement. STaR (Zelikman et al., 2022) bootstraps reasoning by fine-tuning on self-generated chain-of-thought solutions verified against ground truth. Building on this, AdaSTaR (Koh et al., 2025) show that adaptively selecting which solutions to train on, prioritising diversity and calibrating difficulty to model capability, yields substantial efficiency gains. In contrast, OpenSIR requires no annotated data, using self-play for open-ended recursive self-improvement.

## 6 CONCLUSIONS

We present OpenSIR, a self-play framework that enables LLMs to autonomously learn to generate and solve novel problems without external supervision. Starting from only a single trivial math problem, our framework outperforms GRPO-trained models that utilise thousands of human annotations across diverse model families. This approach demonstrates that models can effectively bootstrap mathematical reasoning through recursive self-improvement, eliminating dependence on extensive curated datasets. Our analysis reveals that OpenSIR succeeds by combining difficulty calibration and diversity rewards to create an adaptive curriculum where models continuously discover and master increasingly challenging mathematical concepts. Overall, OpenSIR represents a compelling paradigm for open-ended autonomous mathematical reasoning development, enabling models to recursively expand their capabilities beyond the boundaries of human-annotated data.

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

## A EXTENDED RESULTS AND ANALYSIS

### A.1 FULL RESULTS

We provide the full results of all seeds in Table 5 and 6.

### A.2 CASE STUDY

This section provides further analysis of question-solution pairs during training.

As discussed in Section 4.1, the model generates predominantly invalid problems early in training. Majority of these problems, primarily involve simple mathematical concepts like arithmetic, fail due to missing information (Figures 4 and 5). When attempting complex topics like optimisation, which are rare in the beginning, the model produces problems with missing information and fundamental formulation errors (Figure 6). This reveal the model has limited understanding of underlying mathematical concepts. Invalid problems tend to exhibit low solve rates ($\leq 0.25$) and correspondingly receive lower rewards, helping the model learn to generate valid problems. Consequently, invalid problems decrease rapidly across training (§4.1).

However, not all problems with low solve rates are invalid (§4.2). We find that some problems involving certain topics that are challenging for the model, such as geometric series, persistently exhibit low solve rates (Figures 8. The model struggles with exponentiation calculations, resulting in poor performance on geometric series problems. This reveals a fundamental trade-off in OpenSIR: while higher solve rate thresholds effectively filter out invalid problems, they inevitably discourage exploration of genuinely difficult topics. Since these problems have low solvability scores, they are likely to not receive sufficient encouragement to further explore these topics.

In later training stages, we observe OpenSIR gradually expanding into advanced mathematical domains. After 100 training steps, the model starts to generate problems involving concepts like optimisation (Figure 9), calculus (Figure 10), trigonometry-based physics (Figure 11), probability (Figures 12), among others. While these advanced problems yield lower solve rates, which indicate the model has a limited understanding of these domains, they achieve high novelty scores with large semantic distances and longer solutions. This progression validates how novelty rewards in OpenSIR drive exploration of diverse mathematical concepts, enabling open-ended learning.

> A concert venue charges an admission price per seat and also offers a VIP ticket package that includes food, drinks, and other perks. If the food and other perks are included in the VIP ticket price and are worth \$10 per person, and a group of friends want to buy the same number of VIP tickets as they would with regular tickets, what is the minimum admission price per regular ticket to make the total cost of the VIP tickets equal to or less than the total cost of the regular tickets?

Figure 4: An invalid arithmetic question generated in step 0 with solve rate of 0.25. This question is invalid since the VIP tick price is not provided, and therefore, it's impossible to calculate the minimum regular ticket price.

> Find the percentage difference in the cumulative growth of two continuously compounded interest functions after 5 years: $A = P * e^{(rt)}$, where A is the amount of money accumulated after n years, including interest, P is the principal amount, r is the annual interest rate, and t is the time the money is invested for.

Figure 5: An invalid arithmetic question generated in step 0 with solve rate of 0.125. This question is invalid since the two interest rates and principal amounts are not provided. Hence, it's impossible to calculate the percentage difference with just the general formula provided.

> Consider two positive integers m and n (m $\geq$ n). Given a 2x2 matrix of numbers where each element is a non-negative integer, find the maximum value of the following expression: $ax^2 + by^2 + cxy$, where a, b, and c are constants, subject to the constraint that the sum of any two elements in each row and column of the matrix are equal. What is the maximum possible value of $ax^2 + by^2 + cxy$?

Figure 6: An invalid optimisation question generated in step 0 with solve rate of 0.125. This question is invalid because there are missing information about the constants a, b, and c. There are also ambiguities in the question, such as the role of m and n in the problem. It also did not explain what the elements of the matrix represent. Lastly, it contains problem formulation errors, specifically failing to specify constraints that ensure bounded solutions, demonstrating insufficient understanding of optimization problem structure.

> Consider two positive integers m and n (m $\geq$ n). Given a 2x2 matrix of numbers where each element is a non-negative integer, find the maximum value of the following expression: $ax^2 + by^2 + cxy$, where a, b, and c are constants, subject to the constraint that the sum of any two elements in each row and column of the matrix are equal. What is the maximum possible value of $ax^2 + by^2 + cxy$?

Figure 7: An invalid optimisation question generated in step 0 with solve rate of 0.125. This question is invalid because there are missing information about the constants a, b, and c. There are also ambiguities in the question, such as the role of m and n in the problem. It also did not explain what the elements of the matrix represent. Lastly, it contains problem formulation errors, specifically failing to specify constraints that ensure bounded solutions, demonstrating insufficient understanding of optimization problem structure.

### A.3 FURTHER ANALYSIS ON QUESTIONS DIVERSITY

Figure 13 presents n-gram similarity and concept analysis. We compute ROUGE-L scores between problem texts and extract mathematical concepts using GPT-5 from problems at steps 0, 100, and 200, as well as from the MATH and GSM8K training sets. With diversity rewards (top row), problems maintain low ROUGE-L scores and minimal concept overlap both across training stages and with MATH/GSM8K. Without diversity rewards (bottom row), both textual similarity and concept overlap increase, confirming limited exploration of new problem types.

### A.4 FURTHER ANALYSIS ON QUESTION DIFFICULTY PROGRESSION

Figure 14 compares solve rates between the evolving OpenSIR policy and the fixed instruction model (Base) on problems generated during training. While OpenSIR's solve rate remains stable around 0.7 due to solvability-based problem selection (Section 2.3.1), this constant rate does not imply constant problem difficulty. As OpenSIR improves during training, maintaining the same solve rate requires generating progressively harder problems. To verify this difficulty progression objectively, we measure how the initial instruction model (Base) performs on the same problems. The base model's solve rate first rises (0.48$\rightarrow$0.58 at step 80) then declines (0.58$\rightarrow$0.32 at step 200), confirming the V-shaped difficulty pattern trend from Section 4.1: problems initially become easier as OpenSIR learns appropriate calibration, then progressively harder as it increases challenge. Crucially, this pattern shows that OpenSIR's reasoning ability improved over training.

### A.5 SENSITIVITY TO THE INITIAL SEED PROBLEM

To address whether OpenSIR can robustly escape the limited starting point of a trivial arithmetic seed ("What is 1+1?"), we experiment with two substantially different initial seeds using Llama-3.2-3B-Instruct: a geometry problem from the MATH dataset, representing a different mathematical domain, and a competition-level problem from AIME 2024, which is significantly more challenging than the trivial seed.

**Question**

A company is adding new employees to its workforce. Initially, the company has 500 employees and adds 10 new employees every quarter. If this trend continues for the next 5 years, and each quarter the company adds a fixed percentage increase in the number of new employees (starting with 10%), how many new employees can the company be expected to add during the first 2 years?

**Solution** (Incorrect)

To find the total number of new employees added in the first 2 years, we need to calculate the number of new employees added each quarter for the first 2 years and then sum them up.

First, let's calculate the number of quarters in 2 years: 2 years * 4 quarters/year = 8 quarters.

The company starts with 10 new employees and adds 10% more each quarter. To find the total number of new employees added each quarter, we can use the formula for compound interest:

$$A = P * (1 + r)^n$$

where:
A = total number of new employees after n quarters
P = initial number of new employees (10)
r = percentage increase per quarter (10% or 0.1)
n = number of quarters

However, since we want to find the total number of new employees added each quarter, we need to find the difference between A and the initial number of new employees (10). We can rewrite the formula as:

$$A - P = P * (1 + r)^n$$
$$A - 10 = 10 * (1 + 0.1)^n$$
$$A = 10 * (1.1)^n$$

Now, we can calculate A for each quarter:

Quarter 1 (n = 1): A = $10 * (1.1)^1 = 11$
Quarter 2 (n = 2): A = $10 * (1.1)^2 = 12.1$
Quarter 3 (n = 3): A = $10 * (1.1)^3 = 13.21$
Quarter 4 (n = 4): A = $10 * (1.1)^4 = 14.41$
Quarter 5 (n = 5): A = $10 * (1.1)^5 = 15.71$
Quarter 6 (n = 6): A = $10 * (1.1)^6 = 17.31$
Quarter 7 (n = 7): A = $10 * (1.1)^7 = 19.33$
Quarter 8 (n = 8): A = $10 * (1.1)^8 = 21.59$

Now, we can calculate the total number of new employees added in the first 2 years by summing up the number of new employees added each quarter:

Total = 11 + 12.1 + 13.21 + 14.41 + 15.71 + 17.31 + 19.33 + 21.59
Total = 119.66

Since we can't have a fraction of an employee, we can round down to the nearest whole number.

Therefore, the final answer is: 119

Figure 8: A valid arithmetic problem involving geometric series with a solve rate of 0.125. The solution requires either summing quarterly employee additions or applying the geometric series formula: $a \cdot \frac{r^n - 1}{r - 1}$. Llama-3.2-3B-Instruct struggles with exponentiation calculations, resulting in poor performance on geometric series problems.

A farmer has 10 different types of vegetables to store in a warehouse with 1,000 square meters of floor space. There are four container sizes available:
- Small (5 sq m): Maximum 50 available
- Medium (10 sq m): Maximum 40 available
- Large (15 sq m): Maximum 30 available
- Extra-large (20 sq m): Maximum 25 available
The vegetables have different storage requirements:
- 3 bulky vegetables (pumpkins, watermelons, cabbages) require containers of at least 15 sq m
- 4 medium vegetables (tomatoes, peppers, eggplants, zucchini) require containers of at least 10 sq m
- 3 small vegetables (carrots, onions, potatoes) can fit in any container size
Each vegetable type must be stored in at least one container. What is the maximum number of containers that can be used while satisfying all constraints and not exceeding 1,000 sq m total space?

Figure 9: A valid optimisation problem with a solve rate of 0.375 generated at step 124.

Find the equation of the curve y = f(x) where the derivative is given by $f'(x) = (3x^2 - x - 2)/2x$ and the curve passes through the point (2, 3).

Figure 10: A valid calculus problem with a solve rate of 0.375 generated at step 156.

A golfer hits a ball from the top of a 50-meter high cliff with an initial velocity of 30 m/s at an angle of 45 degrees above the horizontal. What is the horizontal distance traveled by the ball when it hits the ground?

Figure 11: A valid physics problem that involves trigonometry with a solve rate of 0.5 generated at step 172.

Consider a randomly ordered sequence of $n = 3q$ distinct integers $\{a_1, a_2, \ldots, a_{3q}\}$ where $q$ is a positive integer. Define $f$ as the number of adjacent pairs $(a_i, a_{i+1})$ in the sequence where both integers have the same remainder when divided by 3 (i.e., $a_i \bmod 3 = a_{i+1} \bmod 3$). If the integers 1 through $3q$ are randomly permuted to form this sequence, what is the expected value of $f$?

Figure 12: A valid probability problem with a solve rate of 0.25 generated at step 188.

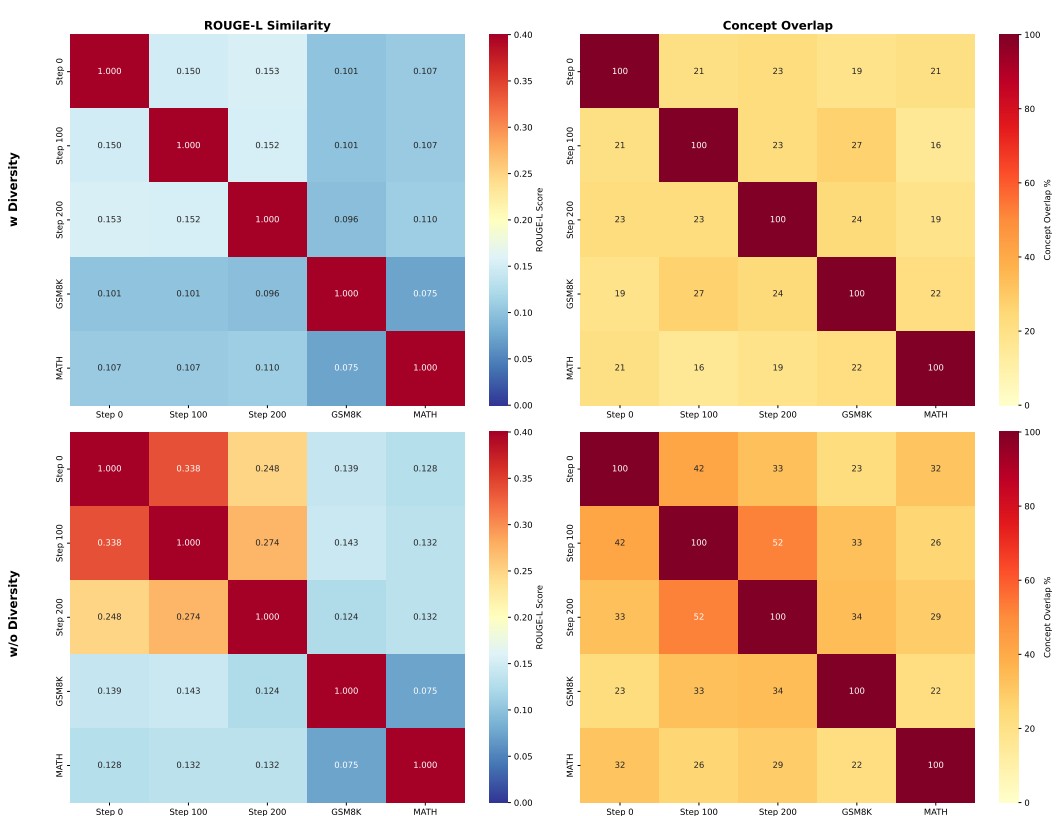

Figure 13: Heatmap visualisation of n-gram similarity (ROUGE-L scores) and concept overlap between generated problems at training steps 0, 100, 200 and reference datasets (MATH, GSM8K). Top row: with diversity reward; Bottom row: without diversity reward. With diversity reward incorporated, the generated problems exhibit low textual similarity and minimal concept overlap, demonstrating effective exploration of diverse problem types.

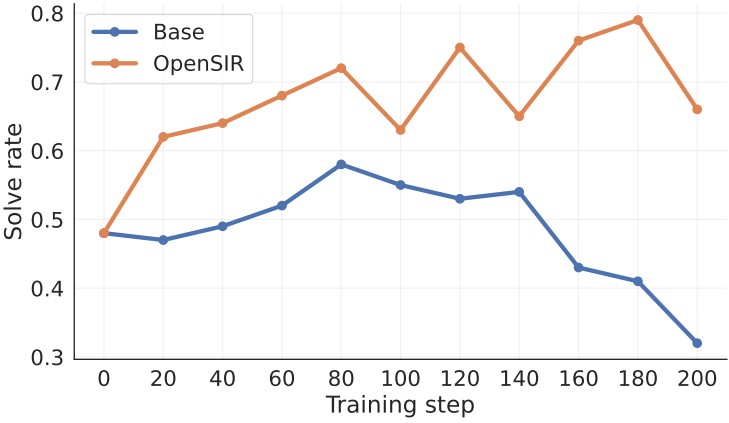

Figure 14: Progression of solve rates of OpenSIR and the initial instruction model as training goes.

$B$ and $C$ trisect $\overline{AD}$ and $M$ is the midpoint of $\overline{AD}$. $MC = 8$. How many units are in the length of $\overline{AD}$?

Figure 15: A geometry problem from the MATH dataset, representing a different mathematical domain from the trivial arithmetic seed.

Every morning Aya goes for a 9-kilometer-long walk and stops at a coffee shop afterwards. When she walks at a constant speed of $s$ kilometers per hour, the walk takes her 4 hours, including $t$ minutes spent in the coffee shop. When she walks $s + 2$ kilometers per hour, the walk takes her 2 hours and 24 minutes, including $t$ minutes spent in the coffee shop. Suppose Aya walks at $s + \frac{1}{2}$ kilometers per hour. Find the number of minutes the walk takes her, including the $t$ minutes spent in the coffee shop.

Figure 16: A competition-level problem from AIME 2024, significantly more challenging than the trivial seed.

Table 7 shows that all three variants achieve nearly identical performance (38.42, 38.67, and 38.81), with differences of less than 0.5 percentage points. This demonstrates that OpenSIR is robust to the initial seed problem, successfully escaping the limited starting point regardless of whether it begins with trivial arithmetic, a different mathematical domain (geometry), or a significantly more challenging competition-level problem.

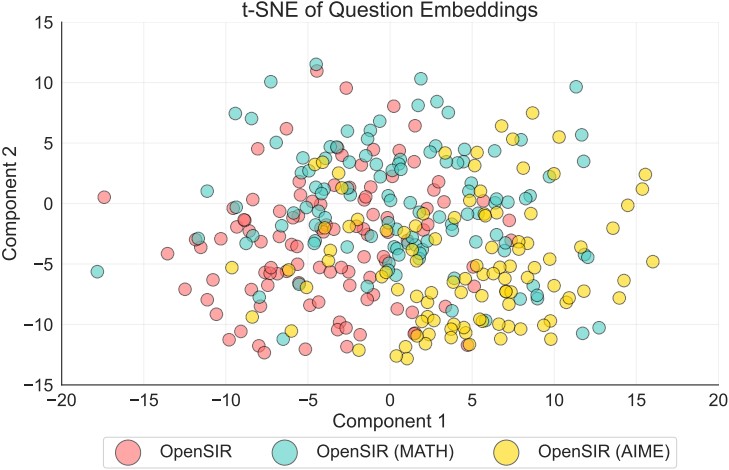

Figure 17: t-SNE visualisation of problem embeddings generated by OpenSIR from three different initial seeds. The substantial overlap demonstrates that the method converges to similar problem distributions regardless of the starting point.

Figure 17 visualises the diversity of problems generated at the final training step across the three different initial seeds. The t-SNE embeddings reveal that all three variants produce diverse problems spanning similar regions of the semantic space, with substantial overlap in their distributions regardless of the initial seed. This confirms that OpenSIR successfully escapes its starting point by exploring a wide range of mathematical concepts, driven by the diversity and solvability rewards that encourage continuous exploration beyond the initial problem domain and difficulty level.

| Models | Seed | GSM8K | MATH-500 | Minerva | College Math | Olympiad-Bench | Avg. |
|---|---|---|---|---|---|---|---|
| **Llama-3.2-3B-Instruct** | | | | | | | |
| Base | - | 73.94 | 42.86 | 15.21 | 28.78 | 13.09 | 34.78 |
| GRPO$_{gsm8k}$ | 42 | 79.60 | 45.41 | 16.34 | 33.31 | 14.71 | 37.87 |
| | 43 | 79.62 | 44.56 | 16.64 | 33.35 | 14.52 | 37.74 |
| | 44 | 79.93 | 45.91 | 15.83 | 33.32 | 14.46 | 37.89 |
| | Avg. | **79.72**$^{\pm0.19}$ | 45.30$^{\pm0.68}$ | 16.27$^{\pm0.41}$ | 33.33$^{\pm0.02}$ | 14.56$^{\pm0.13}$ | 37.83$^{\pm0.37}$ |
| GRPO$_{math}$ | 42 | 76.99 | 45.02 | 16.38 | 33.02 | 14.31 | 37.14 |
| | 43 | 76.51 | 45.23 | 15.95 | 32.87 | 13.85 | 36.88 |
| | 44 | 75.93 | 45.52 | 15.95 | 32.95 | 14.23 | 36.92 |
| | Avg. | 76.48$^{\pm0.53}$ | 45.26$^{\pm0.25}$ | 16.09$^{\pm0.25}$ | 32.95$^{\pm0.07}$ | 14.13$^{\pm0.24}$ | 36.98$^{\pm0.31}$ |
| OpenSIR | 42 | 77.82 | 46.38 | 17.72 | 34.24 | 15.46 | 38.32 |
| | 43 | 78.58 | 45.91 | 17.23 | 34.58 | 15.86 | 38.43 |
| | 44 | 78.43 | 46.38 | 17.44 | 34.45 | 15.84 | 38.51 |
| | Avg. | 78.28$^{\pm0.40}$ | **46.22**$^{\pm0.27}$ | **17.46**$^{\pm0.24}$ | **34.42**$^{\pm0.17}$ | **15.72**$^{\pm0.23}$ | **38.42**$^{\pm0.27}$ |
| **Gemma-2-2B-Instruct** | | | | | | | |
| Base | - | 38.50 | 16.51 | **10.09** | 19.11 | 3.00 | 17.44 |
| GRPO$_{gsm8k}$ | 42 | 58.32 | 18.86 | 7.53 | 20.17 | 3.18 | 21.61 |
| | 43 | 58.86 | 19.21 | 7.96 | 20.77 | 3.08 | 21.98 |
| | 44 | 59.06 | 19.36 | 7.76 | 20.42 | 3.37 | 21.99 |
| | Avg. | **58.75**$^{\pm0.38}$ | 19.14$^{\pm0.26}$ | 7.75$^{\pm0.22}$ | 20.45$^{\pm0.30}$ | 3.21$^{\pm0.15}$ | 21.86$^{\pm0.27}$ |
| GRPO$_{math}$ | 42 | 55.14 | 22.31 | 7.95 | 15.71 | 3.03 | 20.83 |
| | 43 | 53.94 | 22.53 | 7.90 | 15.08 | 3.11 | 20.51 |
| | 44 | 59.01 | 23.44 | 8.02 | 18.15 | 3.57 | 22.44 |
| | Avg. | 56.03$^{\pm2.65}$ | 22.76$^{\pm0.60}$ | 7.96$^{\pm0.06}$ | 16.31$^{\pm1.62}$ | **3.24**$^{\pm0.29}$ | 21.26$^{\pm1.42}$ |
| OpenSIR | 42 | 58.68 | 24.09 | 8.89 | 22.29 | 2.99 | 23.39 |
| | 43 | 58.36 | 25.69 | 10.73 | 26.14 | 3.23 | 24.83 |
| | 44 | 57.03 | 24.49 | 8.89 | 21.66 | 3.24 | 23.06 |
| | Avg. | 58.03$^{\pm0.87}$ | **24.75**$^{\pm0.83}$ | 9.51$^{\pm1.06}$ | **23.36**$^{\pm2.43}$ | 3.15$^{\pm0.14}$ | **23.76**$^{\pm1.30}$ |
| **Qwen-2.5-3B-Instruct** | | | | | | | |
| Base | - | 84.43 | 65.36 | 25.23 | 48.22 | 27.94 | 50.24 |
| GRPO$_{gsm8k}$ | 42 | 84.71 | 65.40 | 26.33 | 48.51 | 28.21 | 50.63 |
| | 43 | 85.16 | 65.80 | 24.84 | 48.46 | 28.50 | 50.55 |
| | 44 | 84.96 | 66.10 | 24.75 | 48.40 | 28.23 | 50.49 |
| | Avg. | 84.94$^{\pm0.23}$ | 65.77$^{\pm0.35}$ | 25.31$^{\pm0.89}$ | 48.46$^{\pm0.06}$ | 28.31$^{\pm0.16}$ | 50.56$^{\pm0.45}$ |
| GRPO$_{math}$ | 42 | 84.24 | 65.74 | 25.23 | 48.53 | 28.23 | 50.39 |
| | 43 | 84.19 | 65.64 | 25.14 | 48.20 | 27.98 | 50.23 |
| | 44 | 84.49 | 66.30 | 24.59 | 48.29 | 28.57 | 50.45 |
| | Avg. | 84.31$^{\pm0.16}$ | **65.89**$^{\pm0.36}$ | 24.98$^{\pm0.35}$ | 48.34$^{\pm0.17}$ | 28.26$^{\pm0.30}$ | 50.36$^{\pm0.28}$ |
| OpenSIR | 42 | 85.43 | 66.17 | 26.49 | 48.88 | 28.86 | 51.17 |
| | 43 | 85.26 | 65.64 | 25.30 | 48.62 | 28.30 | 50.62 |
| | 44 | 85.44 | 65.79 | 26.08 | 48.72 | 27.83 | 50.77 |
| | Avg. | **85.38**$^{\pm0.10}$ | 65.87$^{\pm0.28}$ | **25.96**$^{\pm0.61}$ | **48.74**$^{\pm0.13}$ | **28.33**$^{\pm0.52}$ | **50.85**$^{\pm0.38}$ |

Table 5: Math reasoning evaluation results for 2B/3B models with individual seed reporting. We report avg@16 per problem for each seed (42, 43, 44) and their average with standard deviation as superscript.

| Models | Seed | GSM8K | MATH-500 | Minerva | College Math | Olympiad- Bench | Avg. |
|---|---|---|---|---|---|---|---|
| Llama-3.1-8B-Instruct | | | | | | | |
| Base | - | 84.50 | 47.89 | 22.75 | 34.10 | 16.26 | 41.10 |
| GRPO$_{gsm8k}$ | 42 | 89.73 | 50.89 | 25.61 | 36.16 | 15.65 | 43.55 |
| | 43 | 88.30 | 49.93 | 24.45 | 34.33 | 16.29 | 42.66 |
| | 44 | 88.07 | 50.29 | 24.43 | 34.60 | 17.35 | 42.95 |
| | Avg. | **88.70**$^{\pm 0.73}$ | 50.37$^{\pm 0.39}$ | 24.83$^{\pm 0.55}$ | 35.03$^{\pm 0.81}$ | 16.43$^{\pm 0.70}$ | 43.05$^{\pm 0.62}$ |
| GRPO$_{math}$ | 42 | 86.93 | 51.02 | 24.43 | 35.74 | 17.13 | 43.05 |
| | 43 | 85.98 | 50.53 | 23.88 | 34.48 | 16.56 | 42.29 |
| | 44 | 85.77 | 50.91 | 23.63 | 34.57 | 15.93 | 42.16 |
| | Avg. | 86.23$^{\pm 0.50}$ | 50.82$^{\pm 0.21}$ | 23.98$^{\pm 0.33}$ | 34.93$^{\pm 0.57}$ | 16.54$^{\pm 0.49}$ | 42.50$^{\pm 0.39}$ |
| OpenSIR | 42 | 88.05 | 52.62 | 27.79 | 37.14 | 18.45 | 44.81 |
| | 43 | 87.03 | 52.07 | 27.15 | 35.82 | 17.81 | 43.98 |
| | 44 | 86.82 | 52.45 | 26.94 | 35.91 | 17.18 | 43.86 |
| | Avg. | 87.30$^{\pm 0.54}$ | **52.38**$^{\pm 0.23}$ | **27.29**$^{\pm 0.36}$ | **36.29**$^{\pm 0.60}$ | **17.81**$^{\pm 0.51}$ | **44.21**$^{\pm 0.42}$ |

Table 6: Math reasoning evaluation results for 8B models with individual seed reporting. We report avg@16 per problem for each seed (42, 43, 44) and their average with standard deviation as superscript.

| Model | Acc |
|---|---|
| OpenSIR | 38.42 |
| OpenSIR$_{MATH}$ | 38.67 |
| OpenSIR$_{AIME}$ | 38.81 |

Table 7: Performance of OpenSIR with different initial seed problem.

## A.6 OPENSIR INCENTIVISES REASONING CAPACITY

To verify whether OpenSIR elicits genuine reasoning improvements rather than memorisation, we evaluate pass@k performance on five challenging mathematical benchmarks following (Yue et al., 2025).

Figure 19 shows that OpenSIR consistently outperforms base instruction models across all k values (8–256) on all benchmarks. These results confirm that OpenSIR drives genuine advances in mathematical reasoning capacity.

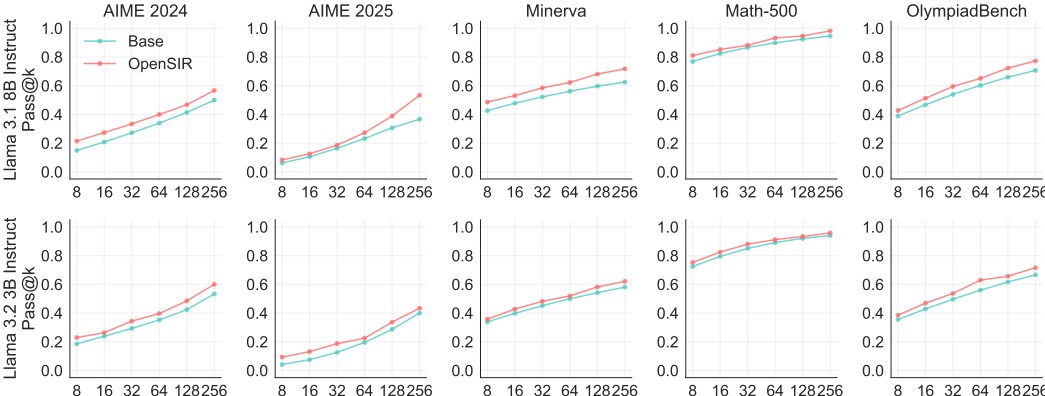

Figure 18: Pass@k curves comparing base instruction models and OpenSIR across five mathematical benchmarks. OpenSIR consistently improves performance across all k values, with stable or increasing gaps at higher k, demonstrating genuine reasoning improvements rather than memorization.

## A.7 PROLONGED TRAINING ANALYSIS

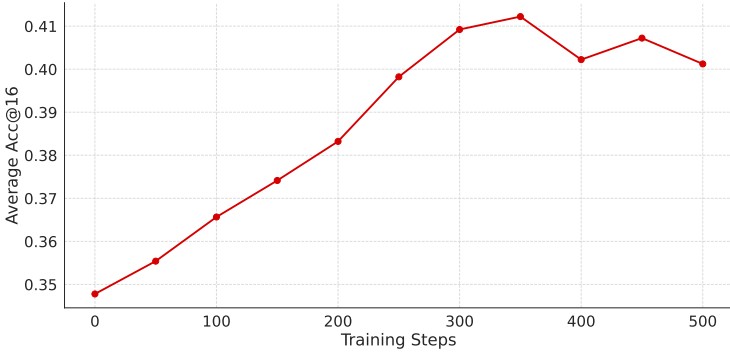

Figure 19: Performance of OpenSIR extended training using Llama-3.2-3B-Instruct.

To better understand the limitation of OpenSIR, we extend training of a single run of Llama-3.2-3B-Instruct to 500 steps, substantially beyond the 200 steps used in the main experiments. Figure 19 shows the evaluation performances over training. We observe consistent improvement from 34.8% to 41.3% at step 350, representing a gain of +6.5 points. Performance then plateaus after step 350 and remains stable around 40-41% through step 500, with no further statistically significant gains.

Preliminary examination of the generated problems suggests a likely cause for this saturation. The model appears to have explored most major mathematical topics by step 350, after which the generated problems become increasingly similar and repetitive. This indicates that the diversity reward mechanism may become less effective over extended training. Future work could investigate more sophisticated diversity mechanisms to foster open-ended exploration over long training horizons.

## A.8 Synergy with Annotated Data

| Model | GSM8K | MATH-500 | Minerva | College Math | OlympiadBench | Avg. |
|-------|-------|----------|---------|--------------|---------------|------|
| Base | 73.94 | 42.86 | 15.21 | 28.78 | 13.09 | 34.78 |
| GRPO$_{gsm8k}$ | 79.72 | 45.30 | 16.27 | 33.33 | 14.56 | 37.83$^{+3.05}$ |
| OpenSIR | 78.28 | 46.22 | 17.46 | 34.42 | 15.72 | 38.42$^{+3.64}$ |
| GSM8K $\rightarrow$ OpenSIR | 81.43 | 46.12 | 19.43 | 36.15 | **18.35** | 40.30$^{+5.52}$ |
| GSM8K & OpenSIR | **81.57** | **49.48** | **20.39** | **36.85** | 18.14 | **41.29**$^{+6.51}$ |

Table 8: The avg@16 performance on five mathematical benchmarks. OpenSIR obtains better results when trained together with GSM8K compared to OpenSIR or GSM8K alone.

While we showed that OpenSIR achieves significant improvements in math reasoning without using annotated data, we further investigate if OpenSIR can be combined with annotated data to achieve even greater performance gains. Having demonstrated that OpenSIR achieves significant improvements without annotated data, we investigate whether combining OpenSIR with annotated data can yield further gains. We focus on Llama-3.2-3B-Instruct and use Gsm8K as the training data, as Table 1 shows that fine-tuning on GSM8K consistently outperforms using MATH.

We explore two training strategies: (1) **GSM8K $\rightarrow$ OpenSIR**: The model is first trained on GSM8K for half the training iterations, then trained with OpenSIR for the remaining half. (2) **GSM8K & OpenSIR**: Each training iteration uses half GSM8k samples and half OpenSIR samples.

Table 8 shows that both setups achieve better performance than using OpenSIR alone or only on GSM8K. One possible explanation for the sequential approach's effectiveness (GSM8K $\rightarrow$ OpenSIR) is that training on GSM8K first may improve the model's foundational reasoning abilities, which could provide a stronger starting point for OpenSIR's self-generated questions. The concurrent approach (GSM8K & OpenSIR) achieves a slight additional edge, which might be attributed to the model receiving better feedback signals for question calibration from the beginning, as it can leverage both supervised and self-generated data simultaneously throughout training. The precise underlying mechanisms for these improvements require further investigation.

## A.9 Computational Cost Analysis

In standard GRPO training, each iteration processes a batch of $B$ prompts, generating $G$ responses for each prompt, resulting in $B \times G$ total forward passes per iteration. In method, each training iteration involves generating $B$ problems and $G$ solution attempts for each problem, yielding $B$ forward passes for problem generation and $B \times G$ forward passes for solution generation, for a total of $B + B \times G = B(1+G)$ forward passes. Compared to the $B \times G$ forward passes in standard GRPO training, method requires an additional $B$ forward passes for problem generation. This represents a relative computational overhead of $\frac{B}{B \times G} = \frac{1}{G}$, or 12.5% with $G = 8$ solution attempts per problem.

Problem embeddings for diversity scoring are computed asynchronously during solution generation, incurring no additional wall-clock time. Cosine distance calculations between problem embeddings require $\mathcal{O}(B \times |\mathcal{P}_t|)$ operations where $|\mathcal{P}_t|$ is the problem pool size, but execute in under 3 seconds per iteration in our experiments—negligible compared to LLM forward passes. Note that while method generates $B$ problems per iteration, only the top-scoring $B/2$ problems are selected for teacher training and another top-scoring $B/2$ for student training, as described in Section 2.4. Overall, method achieves improved performance without human-annotated training data at this modest computational overhead.

# B Annotation Details

One of the authors prepare the samples for annotation, and the rest of the authors annotated the samples with the instructions provide in Figure 20.

You will be presented with multiple sets of 5 math problems to evaluate. For each set, please complete the following three-step annotation process.
# Step 1: Identify Topics
For **each problem**, identify ALL relevant mathematical topics from the following list:
- Algebra
- Geometry
- Calculus
- Probability
- Statistics
- Number Theory
- Combinatorics
- Optimization
- Arithmetic
- Discrete Math
- Trigonometry
# Step 2: Assess Validity
For **each problem**, determine if it is **valid** or **invalid**:
- **Valid**: The problem is logically sound, clearly stated, and can be answered with the given information
- **Invalid**: The problem contains logical flaws, contradictions, insufficient information, or ambiguities that prevent a proper solution
# Step 3: Rank Difficulty
Rank all 5 problems from **easiest to hardest**. Provide your ranking as a sequence of problem numbers.
*Example:* [3, 1, 5, 2, 4] means problem 3 is the easiest and 4 is the hardest.
*Consider these factors when assessing difficulty:*
- Number of steps required
- Complexity of concepts involved
- Level of mathematical knowledge needed
- Computational complexity
# Response Format
Provide your annotations as a JSON list where each element represents one problem set. Here are some examples:

```
[
  {
    "set_id": "SET_1",
    "problems": {
      "1": {"topics": ["Algebra", "Calculus"], "valid": true},
      "2": {"topics": ["Geometry"], "valid": false},
      "3": {"topics": ["Probability"], "valid": true},
      "4": {"topics": ["Number Theory"], "valid": true},
      "5": {"topics": ["Arithmetic"], "valid": true}
    },
    "difficulty_ranking": [5, 3, 1, 2, 4]
  },
  {
    "set_id": "SET_2",
    "problems": {
      "1": {"topics": ["Statistics"], "valid": true},
      "2": {"topics": ["Discrete Math"], "valid": true},
      "3": {"topics": ["Optimization"], "valid": true},
      "4": {"topics": ["Algebra"], "valid": false},
      "5": {"topics": ["Geometry", "Algebra"], "valid": true}
    },
    "difficulty_ranking": [1, 2, 5, 3, 4]
  },
  ...
]
```

Figure 20: The instruction provided to the annotators to annotate problems.

## C  ADDITIONAL ABLATIONS

### C.1  SOLUTION LENGTH REWARD INCREASES PROBLEM COMPLEXITY

| Model | Question Length | Solution Length | Acc |
|---|---|---|---|
| w/ length | 207 | 387 | **38.42** |
| w/o length | 150 | 238 | 37.86 |

Table 9: Comparison of OpenSIR performance with and without solution length reward. Solution length reward improves OpenSIR accuracy and increases average question and solution lengths.

We investigate the impact of the solution length reward in OpenSIR. Table 9 shows this reward improves performance from 37.86% to 38.42%. It also increases the average question length (from 150 to 207 tokens) and solution lengths (from 238 to 387 tokens). By examining the generated questions manually, we find that the policy tends to generate more sophisticated problems involving advanced concepts with this reward, such as linear programming and optimization, which naturally require longer multi-step solutions to solve. These results demonstrate that the solution length reward effectively guides the policy toward generating more complex problems, which in turn leads to better performance.

### C.2  ROBUSTNESS TO DIVERSITY MEASUREMENTS

| Reward | Acc | # Concepts |
|---|---|---|
| Embedding | **38.42** | 5914 |
| Concepts | 38.26 | 6213 |

Table 10: Comparison of diversity measurement approaches in OpenSIR. Despite slight differences in concept coverage, both embedding-based and concept-based diversity rewards yield nearly identical accuracy, demonstrating the framework's robustness to the choice of diversity metric.

We have established the necessity of diversity rewards in Section 4.3. In this section, we further investigate OpenSIR's robustness to different diversity measurement approaches. We implement concept-based diversity by measuring diversity through the mathematical concepts of the problems (Lu et al., 2023; Havrilla et al., 2025). Formally, we define the concept diversity reward as:

$$r_{\text{con}}(q) = \frac{|\mathcal{C}_q| - |\mathcal{C}_q \cap \mathcal{C}_{\mathcal{P}_{t-1}}|}{3} \tag{10}$$

where $\mathcal{C}_q$ are the concepts in problem $q$ and $\mathcal{C}_{\mathcal{P}_{t-1}} = \bigcup_{q' \in \mathcal{P}_{t-1}} \mathcal{C}_{q'}$ represents the union of concepts from all problems in the existing pool. Since each problem contains at most three concepts, this reward calculates the fraction of new concepts introduced.

Table 10 shows that both embedding-based and concept-based diversity rewards achieve similar accuracy (38.42 vs 38.26), demonstrating the framework's robustness to the choice of diversity metric. Beyond accuracy, we examine concept coverage, which refers to the number of unique mathematical concepts discovered during training, as a direct measure of exploratory diversity. As expected, concept-based diversity achieves slightly higher coverage (6,213 concepts) since it explicitly optimises for novel concept discovery. Surprisingly, embedding-based diversity attains comparable coverage (5,914 concepts), 95% of the concept-based approach, despite not tracking concepts explicitly. This suggests that maximising representational spread in embedding space effectively promotes novelty discovery, achieving open-ended learning.

## D  IMPLEMENTATION DETAILS

### D.1  TRAINING DETAILS

We implement OpenSIR based on the TRL framework (von Werra et al., 2020). Table 11 provides a summary of the training hyperparameters used in our experiments.

### D.2  PROMPTS

We detailed the prompt for generating problems in Figure 21 and solving problems in Figure 22.

| Category | Hyperparameter | Value |
|---|---|---|
| Trainer | Learning rate | $3 \times 10^{-7}$ |
| | Optimiser | AdamW (Loshchilov & Hutter, 2018) |
| | Warmup steps | 20 |
| | Training steps | 100/200 |
| | KL loss coefficient | $1 \times 10^{-4}$ |
| | Gradient norm clipping | 0.5 |
| | Seeds | 42/43/44 |
| | GPUs | 3 H100 |
| Rollout | Batch size$^{\dagger}$ | 256 |
| | Max prompt length | 1024 |
| | Max solution length | 2048 |
| | Number of rollouts per prompt | 8 |
| | Temperature | 1.0 |
| Teacher Rewards | Solvability weight ($\alpha$) | 1.0 |
| | Solution length weight ($\lambda$) | 1.0 |
| | Diversity weight ($\gamma$) | 1.0 |
| | Format weight ($\delta$) | 0.1 |
| | Embedding model | Linq-Embed-Mistral (7B) |
| Student Rewards | Accuracy weight | 1.0 |
| | Format weight ($\delta$) | 0.1 |

$^{\dagger}$ The number of rollouts seen for one gradient update.

Table 11: The training configurations for the experiments.

You are given a math problem: {Problem}
Your task is to create a math problem that is conceptually different from the provided problem. The new problem must be answerable with a numerical value or mathematical expression.
First, explain how your new problem differs conceptually from the original problem inside the <think>...</think> tags. Then, present your new problem inside the <problem>...</problem> tags. Finally, identify at most three math concepts required to solve your problem. Provide these concepts in a comma separated list inside the <concepts>...</concepts> tags.

Figure 21: Prompt for generating math problems. {Problem} is a placeholder for the reference problem sampled from the problem pool.

You are a helpful AI Assistant, designed to provide well-reasoned and detailed responses. You FIRST think about the reasoning process step by step and then provide the user with the answer. The last line of your response should be 'Therefore, the final answer is: $\boxed{ANSWER}$' (without quotes) where ANSWER is just the final number or expression that solves the problem.
{Problem}

Figure 22: Prompt for generating solutions to math problems. {Problem} is a placeholder for the actual problem.

