# OpenReview forum: "OpenSIR: Open-Ended Self-Improving Reasoner"
_ICLR.cc/2026/Conference — Submitted to ICLR 2026_

### Official Review · Reviewer_BCw5 · 2025-10-25

**Soundness:** 2
**Presentation:** 2
**Contribution:** 2
**Rating:** 4
**Confidence:** 4

**Summary:**

This paper introduces OpenSIR (Open-Ended Self-Improving Reasoner), a self-play framework designed to enhance LLM reasoning without external supervision by using a single policy that alternates between "teacher" and "student" roles. Starting from a single trivial seed problem , the teacher is trained via reinforcement learning to generate novel problems, optimizing a "novelty" reward that combines both difficulty (calibrated by solve rates and solution length) and diversity (using embedding-based distance to encourage exploration) . The student, in turn, is trained to solve these problems, with correctness determined by majority voting across multiple solution attempts. This self-improving loop allows the model to autonomously bootstrap its capabilities. While the paper has merit, it requires major revision to be ready for publication.

**Strengths:**

The paper is well motivated.

The method is tested on three model families: Llama-3.2-3B-Instruct, Gemma-2-2B-Instruct, Qwen-2.5-3B-Instruct.

Ablation and analysis is appreciated.

**Weaknesses:**

Why is R-Zero not in the baseline? Any other potentially missing baselines?

Pseudocode should be available in the paper. If there is no space in the main body, at least in the appendix. It will dramatically help with clarity and reproducibility.

The work should mention how it compares and contrasts with “AdaSTaR: Adaptive Data Sampling for Training Self-Taught Reasoners” (NeurIPS 2025) as their method also directly tackled the same problem: (1) difficulty (2) diversity. This seems like a key related work.

> To generate novel problems, OpenSIR optimises for both difficulty and diversity, rewarding problems that challenge appropriately while exploring distinct concepts, enabling open-ended mathematical discovery.

There seems to be numerous hyperparameters; e.g. the two s values, alpha, avg@16. What are all the hyperparameters (in the method and also in the empirical study)? How are they set? Heuristically? Empirically? Any sensitivity tests? Large newly introduced hyperparameter space can be a pain for practitioners, especially if the method is sensitive to them.

It seems like this method incurs numerous additional costs. Is the additional cost overhead clearly communicated in the paper? E.g. the embeddings’ cosine similarity will incur costs. This should especially be clearly provided in Tab. 1 where it is very possible that GRPO_gsm8k and GRPO_math has used less training compute than the proposed method.

Does this method only work on the math domain? If so, the scope seems slightly limited.

To my understanding, it is normal practice to keep GRPO going until it hits peak performance. It is concerning that an arbitrary fixed compute budget has been provided.
> To compare models trained on the same number of problem-solution pairs, we train the GRPO baselines with 100 steps, and OpenSIR for 200 steps since OpenSIR allocates half of its training budget to problem generation.

It would be ideal if there was at least one model with larger model size. These models are very small.

Among the three models tested Qwen 2.5 is the strongest one. Naturally, post-training is done on the strongest available base model. Considering this, the Qwen results are most important. However, this method’s gain over baselines in the Qwen results is negligably small; even possibly due to noise. Accuracy gain over best baseline is 0.29% which is virtually no real-world gain.

While it does make sense that the proposed method does not use any labels. There is no need to not use existing labels from e.g. gsm8k, math. There should be experiments with train on existing available labeled train set + opensir to show that this is meaningful in the real-world.
> OpenSIR consistently outperforms GRPO baselines across model architectures despite generating training data through self-play from a single seed problem, while GRPO baselines use over 7,000 human-annotated examples.

**Questions:**

Is there a reason why there is such little use of colors in the text? The clarity of the paper may improve with some coloring, e.g. sections, references.

Does this method work on thinking models, not just instruct models?

---

> ### Author Response · Authors · 2025-11-23
> **Response (1/2)**
>
> $\colorbox{pink}{W1: Missing Baseline - R-Zero}$
>
> We thank the reviewer for this valuable suggestion. We have now incorporated both **R-Zero** and **Absolute Zero** as additional baselines in **Table 1**. **OpenSIR substantially outperforms other self-play methods by 1.75 to 3.38 points** on Llama-3.2-3B-Instruct, Gemma-2-2B-Instruct, and Llama-3.1-8B-Instruct.
>
> $\colorbox{pink}{W2: Pseudocode}$
>
> We thank the reviewer for this comment. We have now included the pseudocode to clarify our method.
>
> $\colorbox{pink}{W3: Comparison with AdaStar}$
>
> We thank the reviewer for bringing up this interesting work. However, **AdaStar is not directly comparable with OpenSIR** since it assumes access to ground-truth solutions to compute difficulty metrics and diversity scores, and selects problems from a fixed dataset during training. For empirical comparison, we focus on methods that share our data-free setting: **R-Zero** and **Absolute Zero**.
>
> $\colorbox{pink}{W4: Hyperparameter Space and Sensitivity}$
>
> For evaluation, it's common to use a moderate temperature for inference and report avg@n for robust evaluation for RL experiments. For simplicity, we follow previous work and use **temp=0.6, top-p=0.95, and n=16** [1].
>
> We report all hyperparameters in training in **Appendix D.1**. These hyperparameters are decided empirically on the baselines with the highest performance, and stick to that for all experiments.
>
> The new hyperparameters in **OpenSIR** compared to typical **GRPO** experiments are the rewards weight, and the solve rate ranges in the solvability reward. We tried changing the reward weights (alpha, gamma, etc.) slightly and found the **performance to be similar**. Therefore, we stick to using **1 for all rewards except the format reward**.
>
> The model is sensitive to the solve rate ranges as mentioned in **Section 4.2**, which controls the tradeoff between the difficulty and validity of the problems, and we find that **[0.5, 0.9] provides the best balance**.
>
> To ensure our hyperparameters are robust, we **validated OpenSIR across three different model families** (Llama, Gemma, Qwen) with **three random seeds each**, showing **consistent improvements**.
>
> Reference:
>
> [1] Liu et al., 2025. SPIRAL: Self-Play on Zero-Sum Games Incentivizes Reasoning via Multi-Agent Multi-Turn Reinforcement Learning. Arxiv preprint.
>
> $\colorbox{pink}{W5: Computational Overhead}$
>
> We thank the reviewer for raising the computational cost concern. We have added a detailed analysis in the new **"Computational Cost Analysis" subsection (Appendix A.9)**.
>
> To clarify the overhead: standard **GRPO** training performs **B×G forward passes** per iteration, where B is the batch size (256) and G is the number of solution attempts per problem (8). **OpenSIR** performs **B×G+B forward passes** per iteration—B forward passes for problem generation plus B×G for solution generation. This represents an additional B forward passes in absolute terms, or a **relative overhead of 12.5%** (B/(B×G) = 1/G). Problem embeddings for diversity scoring are computed asynchronously during solution generation, and cosine distance calculations add **negligible overhead (<3 seconds per iteration)** compared to LLM forward passes.
>
> $\colorbox{pink}{W6: Domain Limitation}$
>
> **OpenSIR works on domains that have objective and verifiable answers**. Our work focuses on the math domain since it's the most extensively studied domain for verifiable answers, which enables comparisons with recent (or concurrent) works [1,2] to showcase **OpenSIR's advantages**. While extensions to other domains with verifiable answers, or even to open-ended domains, would be interesting future work, we believe **OpenSIR represents an important first step** in using self-play to encourage open-ended learning without annotated data.
>
> References:
>
> [1] Zhao et al., 2025. Absolute Zero: Reinforced Self-play Reasoning with Zero Data. The Thirty-ninth Annual Conference on Neural Information Processing Systems.
>
> [2] Huang et al., 2025. R-Zero: Self-Evolving Reasoning LLM from Zero Data. Arxiv Preprint.
>
> $\colorbox{pink}{W7: Extended training with OpenSIR}$
>
>  We thank the reviewer for this important comment. To address this concern, we have added **extended training experiments** with **Llama-3.2-3B-Instruct up to 500 steps**. We observe that performance **peaks at step 350 with an average accuracy of 41.3%**, representing a gain of **+6.5 points** over the base model, and then **plateaus with no further significant improvement through step 500**. For complete details, please refer to **Appendix A.7 (Prolonged Training Analysis)**.

---

> ### Author Response · Authors · 2025-11-23
> **Response (2/2)**
>
> $\colorbox{pink}{W8: Experiments with larger model}$
>
> We thank the reviewer for this suggestion. We have included **Llama-3.1-8B-Instruct** in **Table 1**, which shows that **OpenSIR achieves consistent improvements across different model scales (+3.1 points average for the 8B model)**.
>
> | Model         | GSM8K | MATH-500 | Minerva | College Math | OlympiadBench | Avg.            |
> |---------------|-------|----------|---------|--------------|---------------|-----------------|
> | Base          | 84.50 | 47.89    | 22.75   | 34.10        | 16.26         | 41.10           |
> | GRPO (gsm8k)  | 88.70 | 50.37    | 24.83   | 35.03        | 16.43         | 43.05 (+1.95) |
> | GRPO (math)   | 86.23 | 50.82    | 23.98   | 34.93        | 16.54         | 42.50 (+1.40) |
> | Absolute Zero | 86.89 | 51.38    | 23.21   | 34.39        | 15.96         | 42.37 (+1.27) |
> | R-Zero        | 86.19 | 50.93    | 24.11   | 32.93        | 15.66         | 41.96 (+0.86) |
> | **OpenSIR**   | **87.30** | **52.38** | **27.29** | **36.29** | **17.81** | **44.21 (+3.11)** |
>
> $\colorbox{pink}{W9: Minimal gains on Qwen}$
>
> We thank the reviewer for this important observation. We acknowledge that the limited gains on **Qwen-2.5-3B-Instruct (+0.6 points)** appear negligible. However, as noted in the paper, this aligns with observations of **potential benchmark contamination** in Qwen models. Indeed, **all methods show similarly limited improvements on Qwen**, suggesting the issue might not be unique to OpenSIR.
>
> To address the concern about effectiveness on stronger models, we have included **Llama-3.1-8B-Instruct** in **Table 1**, a significantly more capable model than the 3B variants. **OpenSIR achieves substantial improvements on this larger model (+3.1 points average)**, demonstrating that our method **remains effective on stronger models** while still outperforming GRPO*baselines and other self-play methods.
>
> $\colorbox{pink}{W10: Synergy of OpenSIR with existing labeled data}$
>
> We thank the reviewer for this valuable suggestion. We have added experiments combining **OpenSIR** with existing annotated data in **Appendix A.8 (Synergy with Annotated Data)**. We explore two training strategies using GSM8K: (1) **sequential training (GSM8K followed by OpenSIR)**, and (2) **concurrent training (mixing both in each iteration)**.
>
> | Model           | GSM8K | MATH-500 | Minerva | College Math | OlympiadBench | Avg.            |
> |-----------------|-------|----------|---------|--------------|---------------|-----------------|
> | Base            | 73.94 | 42.86    | 15.21   | 28.78        | 13.09         | 34.78           |
> | GRPO (gsm8k)    | 79.72 | 45.30    | 16.27   | 33.33        | 14.56         | 37.83 (+3.05) |
> | OpenSIR         | 78.28 | 46.22    | 17.46   | 34.42        | 15.72         | 38.42 (+3.64) |
> | GSM8K → OpenSIR | 81.43 | 46.12    | 19.43   | 36.15        | 18.35         | 40.30 (+5.52) |
> | **GSM8K & OpenSIR** | **81.57** | **49.48** | **20.39** | **36.85** | **18.14** | **41.29 (+6.51)** |
>
> The results show that both approaches **achieve better performance than using OpenSIR or GSM8K alone**, with the concurrent approach (**GSM8K & OpenSIR**) achieving the **strongest results at 41.29% average accuracy (+6.51 points** over the initial instruction model, compared to **+3.64 for OpenSIR alone**). This demonstrates that **OpenSIR provides complementary value** to existing annotated data and is indeed **meaningful for real-world applications** where labeled datasets are available.
>
> $\colorbox{pink}{Q1: Use of colors in the text}$
>
> Thank you for the comment. While we agree that using colors might help by highlighting some important parts to improve clarity, we believe that they may be confusing for people with color blindness so we stick to only using colors in our figures.
>
> $\colorbox{pink}{Q2: Experiments with thinking models}$
>
> Thank you for your suggestion. We have further experimented with **DeepSeek-R1-Distill-Llama-8B** and found it also has **significant improvements over the initial model (+3.41)**. This result further validates the **robustness and generality of OpenSIR**.
>
> **DeepSeek-R1-Distill-Llama-8B**
>
> | Model | GSM8K | MATH-500 | Minerva | College Math | OlympiadBench | Avg. |
> | :--- | ---: | ---: | ---: | ---: | ---: | ---: |
> | Base | 71.00 | 71.68 | 27.32 | 45.44 | 39.90 | 51.07 |
> | **OpenSIR** | **78.41** | **74.54** | **29.82** | **48.42** | **41.20** | **54.48** |

---

> ### Comment · Reviewer_BCw5 · 2025-11-24
>
> Many thanks to the authors whom have put significant effort during this phase.
>
> W3. while not needed as a baseline, the methodological motivation overlaps and will be good if discussed in the related work section.
>
> W4. the large and potential sensitive amount of hyperparameters remain a key challenge for practitioners and researchers.
>
> W8. Llama-3.1-8B-Instruct is fairly outdated and rarely used in practice anymore. Why not use at least 3.2 if wanting to show empirical evidence on the llama family?
>
> W9. when a larger model was requested in W8 it was also to see if stronger base model would perform better but again, with the updated ver. qwen 2.5 3b is the best performing model, and this diverges significantly from the model sizes that are used commonly, where it is typical to use at least 7, 8B recent models, like Qwen 2.5, 3 at at least >7B scale and so forth.
>
> Q2. it will be great to see a full experiment on this as this will help resolve some of the issues raised in W8, W9.
>
> While i understand that academia has limited compute, it is important to show that a method would be relevant in real-world practice where significantly more compute is available, and more recent base models are used. There is no need to use Llama 3.1 when much stronger base models exist, like Qwen 3. Also, e.g. if this works at 3B scale but fails to work at every other larger scale the impact of this paper would be highly limited.

---

> ### Author Response · Authors · 2025-11-25
> **Response**
>
> $\colorbox{pink}{W3: Included AdaSTaR in related work}$
>
> Thank you for the suggestion. We have added a paragraph (Synthetic Data) in the related work and mentioned AdaSTAR.
>
> $\colorbox{pink}{W4: Clarification on hyperparameters}$
>
> As mentioned in our previous response, the **only new hyperparameters** introduced by OpenSIR compared to standard GRPO are the **reward weights** and the **solve rate ranges**. We have demonstrated that:
>
> 1. **Reward weights are not sensitive**: We conducted preliminary experiments varying weights (e.g., setting solvability or diversity to 0.5) and found **similar performance across configurations**. Therefore, we use default values of 1 for all reward components but 0.1 for format reward, and we do not expect practitioners to tune these.
>
> 2. **Solve rate ranges are robust**: Our experiments with **[0.5, 0.9]** consistently work well across **diverse model families and sizes** with **three independent random seeds each**. Given this extensive validation, we believe practitioners can confidently adopt these default values without additional tuning.
>
> Regarding the underlying GRPO hyperparameters, while these are shared with standard GRPO training rather than unique to OpenSIR, the growing popularity of GRPO has yielded valuable insights into hyperparameter selection, including recent large-scale studies [1]. By building on a widely adopted RL
> algorithm with established best practices, OpenSIR inherits these practical guidelines, making it accessible to both practitioners and researchers.
>
> In summary, the minimal new hyperparameters introduced are either insensitive (reward weights) or well-validated across diverse settings (solve rate ranges), making OpenSIR **readily accessible to practitioners and researchers**.
>
> Reference:
>
> [1] Khatri et al., 2025. The Art of Scaling Reinforcement Learning Compute for LLMs. arXiv Preprint.
>
>
> $\colorbox{pink}{W8,W9,Q2: Experiment on larger (and thinking) models}$
>
> We thank the reviewer for raising concerns regarding the practical relevance of the experiment results.
>
> We clarify that **Llama-3.1-8B-Instruct is the latest text-only 8B model** in the Llama family.
> Llama 3.2 only offers 1B and 3B text-based instruction models, while the 11B and 90B variants are multimodal models.
> Moreover, Llama 3.3 only offers a 70B text-based model.
> Since our work focuses on text-only models following prior work, **Llama-3.1-8B-Instruct represents the most recent feasible option in the Llama family** within our computational constraints (excluding Llama 3.3 70B).
> However, we acknowledge it may not represent the strongest baseline for demonstrating OpenSIR's potential in real-world applications.
>
> To address this concern, we highlight that our previous response includes results on **DeepSeek-R1-Distill-Llama-8B** , the strongest baseline in our manuscript (51.07 avg. initially), which actually outperforms Qwen-2.5-3B-Instruct (50.24 avg. initially). As reported in our previous response, OpenSIR demonstrates a significant improvement of +3.41 points on this model, validating its effectiveness on a strong 8B-scale model.
>
> Furthermore, we are running additional experiments with **Qwen3-8B**, which is both a stronger model at the 8B scale and a thinking model to address the reviewer's concerns. Results are expected **within 2-3 days** and **will be updated here once available**, providing comprehensive validation across three model families at the 8B scale.

---

> > ### Author Response · Authors · 2025-11-27
> > **Follow-up Response with Larger Model Results**
> >
> > **DeepSeek-R1-Distill-Llama-8B**
> >
> > | Model | GSM8K | MATH-500 | Minerva | College Math | OlympiadBench | Avg. |
> > | :--- | ---: | ---: | ---: | ---: | ---: | ---: |
> > | Base | 71.00 | 71.68 | 27.32 | 45.44 | 39.90 | 51.07 |
> > | GRPO (gsm8k)    | 72.27 | 70.82    | 27.48   | 46.14        | 40.13         | 51.37 (+0.30) |
> > | Absolute Zero | 71.81 | 71.72 | 25.93 | 47.71 | 40.51 | 51.54 (+0.47) |
> > | R-Zero | 75.83 | 72.72 | 25.38 | 45.91 | **41.43** | 52.25 (+1.18) |
> > | **OpenSIR** | **78.41** | **74.54** | **29.82** | **48.42** | 41.20 | **54.48 (+3.41)** |
> >
> > **Qwen3-8B**
> >
> > | Model | GSM8K | MATH-500 | Minerva | College Math | OlympiadBench | Avg. |
> > | :--- | ---: | ---: | ---: | ---: | ---: | ---: |
> > | Base | 95.27 | 79.89 | 38.51 | 50.69 | 42.22 | 61.32 |
> > | GRPO (gsm8k)    | 94.92 | 79.94    | 39.12   | 49.85        | 41.83         | 61.03 (-0.29) |
> > | Absolute Zero | 93.63 | 80.04 | 38.71 | 51.23 | 42.84 | 61.29 (+0.03) |
> > | R-Zero | 94.89 | 81.23 | 39.89 | 52.87 | 42.11 | 62.20 (+0.88) |
> > | **OpenSIR** | **95.34** | **83.13** | **42.28** | **53.07** | **43.77** | **63.52 (+2.20)** |
> >
> > As shown in the tables above, **OpenSIR significantly outperforms all baselines on both models**.  Combined with our Llama-3.1-8B-Instruct results, these experiments demonstrate **consistent gains across three distinct model families at the 8B scale**. We believe this directly addresses the concern regarding whether OpenSIR scales to stronger, more recent models, showcasing its impact and practical relevance for real-world applications.

---

> > > ### Comment · Reviewer_BCw5 · 2025-11-27
> > >
> > > Thanks, I will finalize my score at 6.

---

### Official Review · Reviewer_Jjz1 · 2025-10-26

**Soundness:** 3
**Presentation:** 3
**Contribution:** 2
**Rating:** 4
**Confidence:** 4

**Summary:**

This paper introduces the Open-Ended Self-Improving Reasoner (OpenSIR), a novel framework that enables a Large Language Model (LLM) to autonomously improve its mathematical reasoning capabilities. The core of OpenSIR is a self-play mechanism where a single LLM policy alternates between two roles: a "teacher" that generates new mathematical problems and a "student" that solves them. Starting from a single trivial seed problem (e.g., "What is 1+1?"), the system bootstraps its own learning curriculum without any external human-annotated data. The teacher is rewarded for creating problems that are both appropriately difficult (calibrated via the student's solve rate) and conceptually diverse (measured by embedding distance to previously seen problems). The authors demonstrate that this approach significantly improves the performance of smaller LLMs (Llama-3.2-3B and Gemma-2-2B) on several math reasoning benchmarks, outperforming baselines trained on thousands of human-labeled examples.

**Strengths:**

1.The paper's primary strength lies in its contribution to open-ended, autonomous learning. By successfully demonstrating that an LLM can bootstrap complex reasoning skills from a single trivial example without human supervision, OpenSIR presents a compelling alternative to data-intensive RLHF methods. This addresses a major bottleneck in scaling LLM capabilities and is a significant step towards more autonomous AI systems.

2.The design of the reward function for the teacher role is very effective. Decomposing "novelty" into two intuitive dimensions, difficulty (via scoresol and scorelen) and diversity (scorediv), provides a robust mechanism for generating a dynamic and adaptive curriculum. This allows the model to avoid getting stuck on trivial problems or generating impossibly hard ones, guiding it from basic arithmetic to advanced topics like calculus and trigonometry

**Weaknesses:**

1.The experiments are confined to smaller models (2B-3B parameters). While the results are impressive, the paper shows minimal gains for the stronger Qwen-2.5-3B model. The authors suggest this may be due to benchmark contamination, but it could also indicate that the self-improvement process yields diminishing returns for models that are already highly capable. A discussion on the scalability of this approach to state-of-the-art models (e.g., 8B+) is a notable omission.

2.The self-play loop requires multiple forward passes for each problem generated (G solution attempts per problem) before a single policy update. This process seems computationally expensive compared to standard supervised fine-tuning. The paper does not provide a clear analysis of the computational overhead, making it difficult to assess the practical feasibility and cost-effectiveness of OpenSIR versus simply training on a large, existing dataset.

3.The analysis in Section 4.2 reveals a critical weakness: problems with very low solve rates are often invalid rather than genuinely difficult. The framework's reliance on solve rate as a proxy for difficulty struggles to distinguish between these two cases. While the chosen thresholds (e.g., s_min = 0.5) seem to work, this suggests the curriculum generation might be sensitive to these hyperparameters and could inadvertently filter out challenging but valid new problem domains where the model initially has a very low success rate.

**Questions:**

1. The concept of using self-play for generation and reasoning has been explored in prior work, such as R-Zero. Could the authors further elaborate on the core mechanistic novelty of OpenSIR, particularly its key distinctions from existing approaches? Furthermore, the performance improvement attributed to reinforcement learning appears relatively modest. Does this suggest a potential performance ceiling for this method?

2. The paper's evaluation is primarily conducted on established benchmarks like GSM8K and MATH, where current models already demonstrate strong performance. Does the proposed framework have the potential to generate and solve more complex, competition-level problems (e.g., from AIME)? How robust is the framework's effectiveness in these more challenging scenarios?

3. The experiments are conducted mainly on small-scale models (3B-parameter range). Could the authors comment on the anticipated efficacy of this approach on larger-scale models (e.g., 8B, 14B)? Is the performance upper-bound of the method constrained by the capability ceiling of the initial base model? In other words, is the framework primarily eliciting latent abilities rather than imparting genuinely new skills?

4. The current comparisons are primarily against base models and a general-purpose RL method (GRPO). The paper lacks a direct comparison with contemporary state-of-the-art models in mathematical reasoning  that also leverage synthetic data

---

> ### Author Response · Authors · 2025-11-23
> **Response (1/3)**
>
> $\colorbox{pink}{W1: Experiments on larger models}$
>
> We thank the reviewer's comment on the scalability of OpenSIR. In response to this concern, we have now included experiments with **Llama-3.1-8B-Instruct** in **Table 1**.
>
> | Model         | GSM8K | MATH-500 | Minerva | College Math | OlympiadBench | Avg.            |
> |---------------|-------|----------|---------|--------------|---------------|-----------------|
> | Base          | 84.50 | 47.89    | 22.75   | 34.10        | 16.26         | 41.10           |
> | GRPO (gsm8k)  | 88.70 | 50.37    | 24.83   | 35.03        | 16.43         | 43.05 (+1.95) |
> | GRPO (math)   | 86.23 | 50.82    | 23.98   | 34.93        | 16.54         | 42.50 (+1.40) |
> | Absolute Zero | 86.89 | 51.38    | 23.21   | 34.39        | 15.96         | 42.37 (+1.27) |
> | R-Zero        | 86.19 | 50.93    | 24.11   | 32.93        | 15.66         | 41.96 (+0.86) |
> | **OpenSIR**   | **87.30** | **52.38** | **27.29** | **36.29** | **17.81** | **44.21 (+3.11)** |
>
> **Llama-3.1-8B-Instruct achieves a +3.1 average improvement**, which is comparable to the **+3.6 improvement** observed for Llama-3.2-3B-Instruct with OpenSIR. This suggests that **OpenSIR is able to scale to larger and stronger models**.
>
> $\colorbox{pink}{W2: Computational Overhead}$
>
> We thank the reviewer for raising the computational cost concern. We have added a detailed analysis in the new **"Computational Cost Analysis" subsection (Appendix A.9)**.
>
> To clarify the overhead: standard GRPO training performs **B×G forward passes** per iteration, where B is the batch size (256) and G is the number of solution attempts per problem (8). OpenSIR performs **B×G+B forward passes** per iteration—B forward passes for problem generation plus B×G for solution generation. This represents an additional B forward passes in absolute terms, or a **relative overhead of 12.5%** (B/(B×G) = 1/G). Problem embeddings for diversity scoring are computed asynchronously during solution generation, and cosine distance calculations add **negligible overhead (<3 seconds per iteration)** compared to LLM forward passes.
>
> Importantly, **OpenSIR is explicitly designed for scenarios where models have saturated the available annotated data**. Therefore, OpenSIR is not intended to replace supervised fine-tuning or reinforcement learning, but rather to complement them. When annotated data is exhausted, the relevant comparison becomes "**OpenSIR vs. expensive human annotation for new data**".
>
> $\colorbox{pink}{W3: Difficulty calibration and problem filtering}$
>
> We thank the reviewer for this insightful observation. We acknowledge that OpenSIR's curriculum generation is indeed sensitive to solve rate thresholds, as the framework must balance between identifying genuinely difficult problems and filtering out invalid ones.
>
> However, our chosen thresholds (s_min = 0.5, s_max = 0.9) are designed to greatly minimize the risk of inadvertently filtering out challenging but valid new problem domains. Critically, our analysis in **Section 4.1** demonstrates that **OpenSIR successfully progresses into increasingly complex and diverse mathematical domains** throughout training.
> Moreover, we find that **problem domains initially filtered due to low solve rates are not permanently excluded**—they are adaptively recovered as the model's capabilities improve. For instance, we observed a projectile motion problem filtered at step 8 (solve rate 0.25). This challenging physics problem was initially too difficult for the model. However, at step 172, a similar projectile motion problem reappears and is successfully included (Figure 11). This demonstrates that OpenSIR's curriculum adapts to the model's
> evolving capabilities, **revisiting challenging domains when the model is ready rather than permanently excluding them**.
>
> Furthermore, **OpenSIR incorporates solution length as a complementary mechanism** to capture problem difficulty beyond solve rate alone (**Appendix C.1**). Therefore, we believe that this does not prevent **OpenSIR** from transiting to challenging problem domains.

---

> ### Author Response · Authors · 2025-11-23
> **Response (2/3)**
>
> $\colorbox{pink}{Q1.1: Comparison with R-Zero}$
>
> We thank the reviewer for bringing up a concurrent work (arxiv preprint submitted <2 months before the ICLR deadline). While we are not required to compare against it per the reviewer guidelines (https://iclr.cc/Conferences/2026/ReviewerGuide), we nevertheless included it as a baseline in Table 1, where OpenSIR substantially outperforms R-Zero by 1.75 to 3.38 points across Llama-3.2-3B-Instruct, Gemma-2-2B-Instruct, and Llama-3.1-8B-Instruct. More importantly, our experiments show **R-Zero achieves limited gains on instruction-tune models**, which is precisely the realistic deployment scenario where self-play should excel—when models have already consumed available human-annotated data through instruction tuning.
>
> Moreover, we would like to further elaborate **OpenSIR's core novelties**:
>
> 1. **Diversity-driven open-ended exploration**: OpenSIR explicitly optimises for diversity through embedding-based rewards, enabling exploration from basic arithmetic to advanced topics (calculus, optimization, probability) as shown in **Section 4.1** and **4.3**. **R-Zero** relies solely on repetition penalties without mechanisms to encourage exploration of novel mathematical concepts, which constrains open-ended learning.
>
> 2. **Multi-dimensional difficulty calibration**: Beyond solve rate alone, OpenSIR introduces solution length rewards to complement solvability scoring. Our ablation in **Table 6** shows that it encourages multi-step reasoning problems.
>
> $\colorbox{pink}{Q1.2: Performance ceiling of OpenSIR}$
>
> Thank you for the comment. We have **extended a training run of Llama-3.2-3B-Instruct to 500 steps** from 200 steps in the main experiments. The performance **continued to steadily increase to 41.3% at step 350** from 34.8% initially **(+6.5)**. This suggests that the **performance improvement is significant when given enough training**. Performance then **plateaus after step 350** and remains stable around 40-41% through step 500. For more details, please refer to **Appendix A.7 (Prolonged Training Analysis)**.
>
> $\colorbox{pink}{Q2: Capability on competition-level problems}$
>
> Thank you for your feedback. We further evaluated OpenSIR on AIME 2024/2025 with Llama-3.2-3B-Instruct and Llama-3.1-8B-Instruct. We report the pass@k since these models have very low accuracy on these hard benchmarks.
>
> **Llama-3.2-3B-Instruct**
>
> | Model | aime-2024 pass@16 | aime-2024 pass@64 | aime-2024 pass@256 | aime-2025 pass@16 | aime-2025 pass@64 | aime-2025 pass@256 |
> | :--- | ---: | ---: | ---: | ---: | ---: | ---: |
> | Base | 23.88 | 35.17 | 53.33 | 7.50 | 19.52 | 40.00 |
> | **OpenSIR** | **26.32** | **39.67** | **60.00** | **13.10** | **22.47** | **43.33** |
>
> **Llama-3.1-8B-Instruct**
>
> | Model | aime-2024 pass@16 | aime-2024 pass@64 | aime-2024 pass@256 | aime-2025 pass@16 | aime-2025 pass@64 | aime-2025 pass@256 |
> | :--- | ---: | ---: | ---: | ---: | ---: | ---: |
> | Base | 20.78 | 33.92 | 50.00 | 10.53 | 23.21 | 36.67 |
> | **OpenSIR** | **27.32** | **39.95** | **56.67** | **12.56** | **27.22** | **53.33** |
>
> Results show **significant improvement on wide ranges of k** and demonstrate OpenSIR is effective on these challenging domains.
>
> Moreover, **OpenSIR is able to generate competition-level problems**. An example is provided in **Figure 12**.
>
> $\colorbox{pink}{Q3.1: Scalibility to larger models}$
>
> We thank the reviewer for the feedback. We have now included experiments with **Llama-3.1-8B-Instruct** in **Table 1**.
>
> | Model         | GSM8K | MATH-500 | Minerva | College Math | OlympiadBench | Avg.            |
> |---------------|-------|----------|---------|--------------|---------------|-----------------|
> | Base          | 84.50 | 47.89    | 22.75   | 34.10        | 16.26         | 41.10           |
> | GRPO (gsm8k)  | 88.70 | 50.37    | 24.83   | 35.03        | 16.43         | 43.05 (+1.95) |
> | GRPO (math)   | 86.23 | 50.82    | 23.98   | 34.93        | 16.54         | 42.50 (+1.40) |
> | Absolute Zero | 86.89 | 51.38    | 23.21   | 34.39        | 15.96         | 42.37 (+1.27) |
> | R-Zero        | 86.19 | 50.93    | 24.11   | 32.93        | 15.66         | 41.96 (+0.86) |
> | **OpenSIR**   | **87.30** | **52.38** | **27.29** | **36.29** | **17.81** | **44.21 (+3.11)** |
>
> **Llama-3.1-8B-Instruct achieves a +3.1 average improvement**, which is comparable to the **+3.6 improvement** observed for Llama-3.2-3B-Instruct with OpenSIR. This suggests that **OpenSIR is able to scale to larger and stronger models**.

---

> > ### Author Response · Authors · 2025-11-23
> > **Response (3/3)**
> >
> > $\colorbox{pink}{Q3.2: Eliciting latent abilities or imparting genuinely new skills}$
> >
> > Thank you for raising this insightful question. We follow previous work [1] to compare the **pass@k performance** of OpenSIR and the initial model (Details in **Appendix A.6, Figure 18**). Results show that **OpenSIR consistently improves pass@k across all k values** on five challenging benchmarks over the initial model. This suggests that **OpenSIR is imparting genuinely new skills**.
> >
> > Reference:
> >
> > [1] Yue et al., 2025. Does Reinforcement Learning Really Incentivize Reasoning Capacity in LLMs Beyond the Base Model? NeurIPS 2025.
> >
> > $\colorbox{pink}{Q4: Comparison with state-of-the-art methods}$
> >
> > We have now included comparisons to **Absolute Zero** and **R-Zero** in **Table 1**. These two methods are recent (or concurrent) works that also use self-play to construct synthetic data to improve LLM reasoning without using annotated data. We have outlined the difference between OpenSIR and these two works.
> > Results show that **OpenSIR significantly outperforms these two methods** across all models.

---

> ### Comment · Reviewer_Jjz1 · 2025-11-26
>
> Thank you for the authors' detailed response, which has addressed some of my concerns. However, I still have the following questions:
>
> 1. Regarding W2: How many seed problems are included in each batch, one or multiple? If there is only one, wouldn't this lack diversity for RL training?
>
> 2. Could you provide a performance comparison between OpenSIR,DAPO,Absolute Zero and R-Zero on benchmarks such as AIME24 and AIME25 using Llama-3.1-8B-Instruct?
>
> 3. Could you highlight the unique contributions of your work compared to Absolute Zero? The two approaches appear conceptually similar.

---

> > ### Author Response · Authors · 2025-11-27
> > **Response**
> >
> > Thank you for your response, and we are glad that we have addressed some of your concerns. Please see our response below for your follow-up questions.
> >
> > $\colorbox{pink}{Clarification on seed problem}$
> >
> > We only use **one seed problem** to initialise the problem pool. Note that the seed problem is only directly relevant in the **first iteration**. In each subsequent iteration, OpenSIR samples a fixed number of reference questions from the growing problem pool that accumulates all valid questions generated so far.
> >
> > Even in the first iteration, when the problem pool contains only the seed question, the generation prompt (Figure 21) explicitly instructs the model to creates problems that are **conceptually different from the provided reference**. Combined with moderate sampling temperature, this samples questions that are fairly diverse. We provide some examples for questions generated at the beginning:
> >
> >
> >
> > ```
> > A bag contains 24 marbles, of which 1/4 are red, 1/3 are blue, and the remaining marbles are green. If a marble is randomly selected from the bag, what is the probability that it is either red or green? Express your answer as a percentage.
> > ```
> >
> > ```
> > If a movie starts at 3:45 PM and lasts for 1 hour and 25 minutes, what time does the movie end?
> > ```
> >
> > During training, the diversity reward encourages the model to generate varied questions throughout training, as analysed in Section 4.1 and 4.3.
> >
> > $\colorbox{pink}{Further results on AIME with Llama-3.1-8B-Instruct}$
> >
> > Thank you for your valuable suggestion. We believe you may be referring to the GRPO baselines in our paper (rather than DAPO). We have conducted the requested comparison and present the results below.
> >
> > **Performance on AIME (Llama-3.1-8B-Instruct)**
> >
> > | Model | aime-2024 pass@16 | aime-2024 pass@64 | aime-2024 pass@256 | aime-2025 pass@16 | aime-2025 pass@64 | aime-2025 pass@256 |
> > | :--- | ---: | ---: | ---: | ---: | ---: | ---: |
> > | Base | 20.78 | 33.92 | 50.00 | 10.53 | 23.21 | 36.67 |
> > | GRPO (gsm8k) | 24.64 | 36.25 | 46.67 | 11.43 | 25.84 | 33.33
> > | GRPO (math) | 22.93 | 35.84 | 43.33 | 10.82 | 24.93 | 30.00
> > | Absolute Zero | 21.87 | 34.67 | 36.67 | 10.84 | 24.18 | 30.00 |
> > | R-Zero | 22.29 | 35.72 | 40.00 | 10.73 | 25.71 | 33.33 |
> > | **OpenSIR** | **27.32** | **39.95** | **56.67** | **12.56** | **27.22** | **53.33** |
> >
> >
> > The results shows that OpenSIR significantly outperforms all baselines on competition-level math benchmarks using Llama-3.1-8B-Instruct.
> >
> > Notably, **OpenSIR is the only method that improves pass@256 over the initial model**, implying OpenSIR  is the only method that incentivises genuine reasoning improvements.
> >
> > $\colorbox{pink}{Comparison with Absolute Zero}$
> >
> > Thank you for the question. While OpenSIR shares the high-level goal of bootstrapping questions with self-play to self-improve without human annotations, there are **two fundamental differences** that distinguish our approach:
> >
> > 1. **Verifier-free design to generalises beyond coding**
> >
> > Absolute Zero relies on Python execution to verify both generated problems and their solutions, restricting it to deterministic coding tasks where ground-truth answers can be programmatically checked. This dependency inherently limits the diversity of problems that can be generated and constrains generalisability. OpenSIR, in contrast, only uses majority voting **without an external verifier**, leveraging solve rate thresholds and solution length to ensure quality and capture difficulty.
> >
> >
> > 2. **Diversity-driven Open-ended learning**
> >
> > Absolute Zero optimises for problem difficulty through minimal solve rates, but **lacks explicit mechanisms for encouraging diversity or driving open-ended exploration.** OpenSIR, by contrast, explicitly optimises for diversity through embedding-based rewards that **encourage the model to explore varied mathematical concepts**. This diversity-driven objective is central to genuine open-ended learning. Our ablation study in Section 4.4 shows its importance in the performance.
> >
> > Our results demonstrate that OpenSIR consistently outperforms Absolute Zero. We believe these gains reflect the richer, more diverse training curriculum that emerges from removing verifier constraints and explicitly rewarding conceptual variety.

---

### Official Review · Reviewer_w5VH · 2025-11-01

**Soundness:** 2
**Presentation:** 2
**Contribution:** 2
**Rating:** 4
**Confidence:** 4

**Summary:**

This paper introduces OpenSIR (Open-Ended Self-Improving Reasoner), a self-play framework for large language models (LLMs) that autonomously improves reasoning abilities without external supervision. OpenSIR alternates teacher and student roles, generating and solving novel problems optimized for difficulty and diversity, enabling open-ended mathematical discovery. Starting from a single trivial seed problem, OpenSIR drives autonomous progression from basic to advanced concepts. Experiments show significant performance improvements on benchmarks like GSM8K and College Math, with models achieving substantial gains. The framework's adaptive teacher-student co-evolution fosters diverse exploration and calibrated learning, advancing LLM reasoning capabilities effectively.

**Strengths:**

1. The paper tests its framework, OpenSIR, across multiple benchmarks (e.g., GSM8K, College Math) using various backbone LLMs, such as Llama-3.2B-Instruct and Gemma-2-2B-Instruct. This demonstrates some generality and effectiveness of the approach across different models and tasks.

2. The paper tackles an important topic—using reinforcement learning (RL) to improve LLM reasoning capabilities. RL is a compelling approach for driving autonomous learning, making this work relevant and interesting for advancing LLMs.

**Weaknesses:**

1. The core idea of OpenSIR lacks novelty, appearing more like a combination of popular concepts (self-play, RL, curriculum learning) rather than introducing a new approach. The paper could benefit from showcasing deeper insights or unique contributions that distinguish it from existing methods.

2.  The authors do not provide code or other necessary materials, making it difficult for researchers to replicate the results or experiment with the framework. Including well-documented code and resources would significantly enhance the paper's impact and accessibility.

3. The font used in Figure 1 appears informal and less readable, which detracts from the professional presentation of the paper. Using a more formal and easily readable font would improve the clarity and visual impact of the figure, making it more suitable for academic audiences.

4. The experiments are conducted on relatively small models. While the results are promising, the robustness and effectiveness of the proposed method on larger-scale models remain unverified. Expanding the experiments to larger models would strengthen the paper's claims and demonstrate broader applicability.

**Questions:**

Refer to Weaknesses

---

> ### Author Response · Authors · 2025-11-23
> **Response**
>
> $\colorbox{pink}{W1: Novelty of OpenSIR}$
>
> Thank you for the comment. Our work has **two main novelties** that distinguish it from all prior self-play methods:
>
> 1. **We are the first to incorporate diversity rewards** in a self-play framework for math reasoning to achieve **open-ended learning**, which is a critical capability absent in prior or concurrent work [1,2]. Our analysis shows that it enables the model to **discover and explore varied mathematical concepts**, from basic arithmetic to advanced mathematics—including calculus, optimisation, trigonometry, and probability.
>
> 2. **We are the first to systematically evaluate self-play** for mathematical reasoning on **instruction models** (rather than base models) with **multiple random seeds**, reflecting realistic deployment scenarios where models have already consumed available annotated data. While Absolute Zero and R-Zero (two recent/concurrent works) work on base models, our experiments show they **provide minimal gains on instruction-tuned models**. Our work has **practical implications for scaling LLM reasoning capabilities** when annotated data is exhausted.
>
> References:
>
> [1] Zhao et al., 2025. Absolute Zero: Reinforced Self-play Reasoning with Zero Data. The Thirty-ninth Annual Conference on Neural Information Processing Systems.
>
> [2] Huang et al., 2025. R-Zero: Self-Evolving Reasoning LLM from Zero Data. Arxiv Preprint.
>
> $\colorbox{pink}{W2: Code}$
>
> Thank you for the reminder. We have attached the code in the submission. We will also open source the code, model checkpoints, etc. after the discussion period.
>
> $\colorbox{pink}{W3: Main figure}$
>
> Thank you for the comment. We have changed the figure into a more formal one.
>
> $\colorbox{pink}{W4: Experiments on Larger Models}$
>
> Thank you for the feedback. We have included experiments with **Llama-3.1-8B-Instruct** in **Table 1**.
>
> | Model         | GSM8K | MATH-500 | Minerva | College Math | OlympiadBench | Avg.            |
> |---------------|-------|----------|---------|--------------|---------------|-----------------|
> | Base          | 84.50 | 47.89    | 22.75   | 34.10        | 16.26         | 41.10           |
> | GRPO (gsm8k)  | 88.70 | 50.37    | 24.83   | 35.03        | 16.43         | 43.05 (+1.95) |
> | GRPO (math)   | 86.23 | 50.82    | 23.98   | 34.93        | 16.54         | 42.50 (+1.40) |
> | Absolute Zero | 86.89 | 51.38    | 23.21   | 34.39        | 15.96         | 42.37 (+1.27) |
> | R-Zero        | 86.19 | 50.93    | 24.11   | 32.93        | 15.66         | 41.96 (+0.86) |
> | **OpenSIR**   | **87.30** | **52.38** | **27.29** | **36.29** | **17.81** | **44.21 (+3.11)** |
>
> Results show that **OpenSIR achieves a +3.1 average improvement** in **Llama-3.1-8B-Instruct**, which is comparable to the **+3.6 improvement** observed for Llama-3.2-3B-Instruct.

---

> > ### Comment · Reviewer_w5VH · 2025-11-26
> >
> > Thank your for the response. I considered the other reviewers' suggestions and decided to keep the score.

---

> > > ### Author Response · Authors · 2025-11-27
> > > **Response**
> > >
> > > Thank you for the response.  We have put significant effort into addressing all reviewers' concerns through additional experiments and analyses. If there are any concerns that remain unresolved, we are happy to provide further information to better represent the contribution of OpenSIR.

---

### Official Review · Reviewer_LBsx · 2025-11-02

**Soundness:** 3
**Presentation:** 3
**Contribution:** 3
**Rating:** 4
**Confidence:** 3

**Summary:**

This paper introduces OpenSIR, a self-play reinforcement learning framework where a single policy jointly optimizes two roles: a teacher that generates novel and diverse math problems, and a student that solves them accurately. Both roles are jointly trained to form an open-ended self-improvement loop that enables the model to continuously enhance its problem generation and reasoning abilities. This dual-role optimization enables the model to bootstrap from a single seed problem and progressively enhance both problem generation and reasoning capability without human supervision. Experiments show that OpenSIR consistently outperforms GRPO baselines and base instruction models, achieving comparable or superior accuracy without any human-annotated data.

**Strengths:**

- The proposed approach significantly outperforms supervised RL approaches (GRPO) and instruction-tuned baselines across a number of models and benchmarks.

- The proposed approach requires no human-annotated data, reducing cost and reliance on manual labeling.

- Joint optimization of teacher and student creates a self-calibrating cycle, enabling continuous self-generated training at optimal difficulty.

**Weaknesses:**

- In 4.1 Figure 2, the observed V-shaped difficulty trend is interesting, but the authors should provide evidence of the student model’s performance over training (e.g., accuracy or solve rate) to substantiate the claim that this pattern reflects true self-calibration.

- In 2.1, the author states, “We initialise the problem pool P_0 with a single trivial problem (“What is 1+1?”)”  Given the simplicity of this seed, it is worth discussing whether and how this choice constrains the initial diversity or attainable difficulty of the generated problems, and whether the model can robustly escape such a limited starting point.

- While the paper conducts ablations on diversity and length rewards and dual-role training, it does not provide individual analyses for the solvability reward components. It remains unclear how the component contributes to the overall accuracy performance.

**Questions:**

- In Table 8, equal weights (α = λ = γ = 1.0, δ = 0.1) are assigned to most teacher rewards and 1.0 to the student accuracy reward. Could the authors clarify how these weights were chosen? Were they empirically tuned, or are the results robust to moderate changes in these hyperparameters?

---

> ### Author Response · Authors · 2025-11-23
> **Response**
>
> $\colorbox{pink}{W1: Additional evidence for the V-shaped difficulty trend}$
>
> Thank you for your feedback. We added **Appendix A.4**, which compares the solve rates between the evolving OpenSIR policy and the initial instruction model on problems generated during training. Since we prioritise problems with a solve rate near 0.7, OpenSIR's solve rate hovers around 0.7 as intended. While OpenSIR's solve rate remains stable around 0.7 by design (solvability-based selection), this constant rate does not imply a constant level of difficulty.
>
> We observe that the initial instruction model (Base) exhibits the **V-shaped difficulty trend**: the solve rate first rises, then declines. This **V-shaped trend confirms that problems initially become easier** as OpenSIR learns appropriate difficulty calibration, then **progressively harder** as OpenSIR's solving capability improves.
>
>   $\colorbox{pink}{W2: Different initial seed problem}$
>
> Thank you for this valuable comment. We have added experiments with two different seeds in **Appendix A.5**: a geometry problem from MATH (different domain) and a competition-level AIME 2024 problem (significantly harder).
>
> | Model               | Acc   |
> |---------------------|-------|
> | OpenSIR             | 38.42 |
> | OpenSIR (MATH seed) | 38.67 |
> | OpenSIR (AIME seed) | 38.81 |
>
> **All three seeds achieve nearly identical performance** and produce **similar diversity** in problem distributions (visualized via t-SNE, See Figure 17). This demonstrates that **OpenSIR robustly escapes the limited starting point**, driven by diversity and solvability rewards.
>
>   $\colorbox{pink}{W3: Solvability reward analysis}$
>
> We would like to clarify that **Section 4.2 ("Difficulty-Validity Trade-off")** provides an ablation study specifically analyzing the solvability reward component by examining different solve-rate threshold configurations.
>
> Our analysis reveals that **lowering the solve-rate threshold from 0.5 to 0.1** produces moderately harder problems (GPT-5 solve rate: **89.82%→78.31%**) but causes **validity to plummet (70.82%→42.31%)**, indicating that very low solve rates signal malformed problems rather than genuine difficulty. Critically, **model performance consistently degrades with lower thresholds**, empirically validating the solvability reward's contribution to overall accuracy by ensuring appropriately challenging yet well-formed training problems.
>
>   $\colorbox{pink}{Q1: How were the hyperparameters chosen}$
>
> Thank you for this question. We conducted preliminary experiments where we set either the solvability or diversity reward weight to 0.5 while keeping other weights unchanged. These experiments showed **similar performance across configurations**, indicating **robustness to moderate variations** in relative weighting.
>
> Based on these findings, we adopted **uniform weights (α = λ = γ = 1.0) for simplicity**, with format reward **δ = 0.1** since the instruction models can mostly follow the instructions in the beginning.

---

### Author Response · Authors · 2025-11-23
**Summary of Updates/Responses**

We sincerely thank all reviewers for their thoughtful feedback and comments. We have updated our manuscript, including multiple new experiments and analyses to address all the raised concerns. All changes in the manuscript are highlighted in blue. Below, we summarise the key additions

* **Comparisons with Absolute Zero and R-Zero (Table 1\)**: We add direct comparisons to two recent self-play methods for improving mathematical reasoning without annotated data. Results demonstrate that **OpenSIR substantially outperforms both approaches**.   (Addresses missing baseline and SOTA comparisons)
* **Results on 8B-scale methods**: Added experiments on Llama-3.1-8B-Instruct (+3.1 points), DeepSeek-R1-Distill-Llama-8B (+3.41 points), and Qwen3-8B (+2.20 points). Results demonstrate that **OpenSIR scales effectively to stronger, larger models**. (Addresses scalability on larger, stronger frontier models)
* **Pseudocode (Algorithm 1\)**: Added comprehensive pseudocode to clarify the complete OpenSIR training procedure.
* **Further analysis on question difficulty progression (Appendix A.4)**: Tracks how the model's solve rate evolves on the generated problems, providing additional evidence to further support the V-shaped difficulty trend.
* **Experiments with different initial seed problems (Appendix A.5)**: Tested with a MATH geometry problem and an AIME 2024 competition problem. Results show similar performance and comparable diversity in generated problems, demonstrating robustness to initialisation.
* **Pass@k experiments (Appendix A.6)**: Evaluated across k=8 to 256 on five math benchmarks, demonstrating that OpenSIR drives genuine improvements in mathematical reasoning capacity. (Addresses questions on whether OpenSIR imparts new skills versus eliciting latent abilities)
* **Extended training from 200 to 500 steps (Appendix A.7)**: Results demonstrate that OpenSIR scales effectively, achieving consistent improvements up to step 350 (+6.5 points total).
(Addresses concerns on the limited improvements in the main experiments and the potential ceilings of OpenSIR)
* **Experiments combining OpenSIR with annotated data (Appendix A.8)**: Results show that OpenSIR achieves higher performance (+6.51 points) when combined with GSM8K compared to either approach alone, demonstrating complementary value.
* **Computational cost analysis (Appendix A.9)**: Shows that the computational overhead is minimal (only 12.5% compared to standard GRPO training), achieving substantial improvements without human annotations at modest additional cost. (Addresses feasibility and compute-overhead concerns)

We sincerely thank the reviewers for all the feedback that has made the revised manuscript substantially stronger. Taken together, the new experiments demonstrate that OpenSIR is robust across different model sizes, families, and initialisation choices, and that it consistently yields genuine reasoning improvements at modest additional compute cost, and shows substantial room for further scaling at minimal additional computational overhead. We believe the revision now clearly establishes OpenSIR's advantages relative to existing self-play approaches and provides sufficient empirical evidence to support its main claims.

---

### Meta-Review · Area_Chair_c4Xa · 2025-12-23

**Summary:**

This paper proposes a self-play reinforcement learning framework named OpenSIR. Four reviewers submitted evaluation comments, to which the authors provided rebuttals, and most reviewers participated in the discussion. The final score approached the borderline, with three reviewers leaning toward rejection. After thoroughly reviewing the paper and the authors' responses, the AC noted that some reviewer concerns remained unresolved, such as the novelty of the method and the need for more comprehensive experiments and evaluations. The AC recommends that the authors revise and resubmit to a subsequent conference.

**Reviewer Concerns:**

Experimental and detail-related issues have been resolved, but questions regarding the novelty of the methodology remain difficult to address.

**Reviewer Scores:**

Most reviewers participated in the discussion, and reviewer BCw5's score will be raised to 6 points. I believe the scores for the other reviewers will remain unchanged.

---

### Decision · Program_Chairs · 2026-01-26

Reject